# Nitro-Deficient Niclosamide Confers Reduced Genotoxicity and Retains Mitochondrial Uncoupling Activity for Cancer Therapy

**DOI:** 10.3390/ijms221910420

**Published:** 2021-09-27

**Authors:** Tsz Wai Ngai, Gamal Ahmed Elfar, Pearlyn Yeo, Nicholas Phua, Jin Hui Hor, Shuwen Chen, Ying Swan Ho, Chit Fang Cheok

**Affiliations:** 1Institute of Molecular and Cell Biology, Agency for Science, Technology and Research, Singapore 138673, Singapore; tszwn@imcb.a-star.edu.sg (T.W.N.); gamalar@imcb.a-star.edu.sg (G.A.E.); Pearlyn_Yeo@imcb.a-star.edu.sg (P.Y.); pssnchls@gmail.com (N.P.); jhhor@imcb.a-star.edu.sg (J.H.H.); 2Department of Pathology, Yong Loo Lin School of Medicine, National University of Singapore, Singapore 119074, Singapore; 3Analytical Science and Technology (Metabolomics), Bioprocessing Technology Institute, 20 Biopolis Way, Centros #06-01, Singapore 138668, Singapore; chen_shuwen@bti.a-star.edu.sg (S.C.); ho_ying_swan@bti.a-star.edu.sg (Y.S.H.)

**Keywords:** niclosamide, genotoxicity, mitochondrial uncoupling, p53, lipidomics

## Abstract

Niclosamide is an oral anthelmintic drug, approved for use against tapeworm infections. Recent studies suggest however that niclosamide may have broader clinical applications in cancers, spurring increased interest in the functions and mechanisms of niclosamide. Previously, we reported that niclosamide targets a metabolic vulnerability in p53-deficient tumours, providing a basis for patient stratification and personalised treatment strategies. In the present study, we functionally characterised the contribution of the aniline 4′-NO_2_ group on niclosamide to its cellular activities. We demonstrated that niclosamide induces genome-wide DNA damage that is mechanistically uncoupled from its antitumour effects mediated through mitochondrial uncoupling. Elimination of the nitro group in ND-Nic analogue significantly reduced γH2AX signals and DNA breaks while preserving its antitumour mechanism mediated through a calcium signalling pathway and arachidonic acid metabolism. Lipidomics profiling further revealed that ND-Nic-treated cells retained a metabolite profile characteristic of niclosamide-treated cells. Notably, quantitative scoring of drug sensitivity suggests that elimination of its nitro group enhanced the target selectivity of niclosamide against p53 deficiency. Importantly, the results also raise concern that niclosamide may impose a pleiotropic genotoxic effect, which limits its clinical efficacy and warrants further investigation into alternative drug analogues that may ameliorate any potential unwanted side effects.

## 1. Introduction

Niclosamide, 5-chloro-N-(2-chloro-4-nitrophenyl)-2-hydroxybenzamide, is an oral anthelmintic drug approved for the treatment of various tapeworm infections in humans [1,2]. Mechanistically, niclosamide functions as a protonophore and disrupts parasitic mitochondrial function via the uncoupling of oxidative phosphorylation [3,4,5]. A number of mitochondrial uncouplers have emerged as promising anti-cancer agents with selective anti-cancer effects, pointing to the potential of exploiting metabolic and mitochondrial vulnerabilities in cancers [6,7,8]. Recently, studies revealed that niclosamide elicits anti-neoplastic effects against cancer cell lines across broad cancer types including acute myelogenous leukaemia (AML) stem cells, lung, brain, ovarian, breast, and colorectal cancer [9,10,11,12,13,14,15,16] and in patient-derived xenografts and organoids [17,18]. This unravels the potential of re-purposing niclosamide as an antitumour agent, an attractive possibility given that niclosamide is Food and Drug Administration (FDA) approved with an established safety and pharmacokinetics profile. Niclosamide is currently in clinical trials against tumours in advanced and recurrent stages [19,20,21].

While its action as a mild mitochondrial uncoupler is sufficient to kill tapeworms in the gastrointestinal (GI) tract at a single dosage [22,23], however, repurposing it as a chemotherapeutic modality inevitably involves optimising drug dosage to maximal tolerable limits to achieve clinical efficacy. Further clinical development of niclosamide requires investigation into the potential adverse effects that may accompany changes in the drug dose regimens even though the safety and pharmacokinetics of niclosamide as an antiparasitic drug at single dose is already established. In addition, functional dissection of the molecular features of niclosamide will allow for a better characterisation of its structure-activity relationship (SAR), which will greatly aid in drug design. Ideally, the functional uncoupling between the defined biological activity of a drug from any pleiotropic effects or nonspecific toxicity in structure-activity experiments would provide an optimal scenario for drug design, with future development focused on reducing or eliminating nonspecific drug action.

The structure of niclosamide comprises its salicylic acid and aniline motifs. The hydroxyl group on the salicylic acid ring motif of niclosamide is predicted to be critical for its mitochondrial uncoupling function as well as its defined antitumour activities [17,24]. By undergoing reversible protonation, niclosamide acts as an ionophore to translocate protons across the inner mitochondrial membrane (IMM) and dissipate the proton gradient across the IMM, leading to the uncoupling of nutrient oxidation from ATP generation. The depolarisation of mitochondrial membrane is a characteristic of protonophores, including niclosamide and classical uncouplers such as carbonyl cyanide p-(tri-fluromethoxy)phenyl-hydrazone (FCCP) and 2,4-dinitrophenol. Substituting the hydroxyl group on the salicylic acid motif of niclosamide with a 2-methoxy substituent completely abrogates its ability to uncouple mitochondrial respiration [25]. Whereas the aniline 4′-NO_2_ group may undergo nitroreduction yielding anilines (aromatic amines), which may be potentially mutagenic [26,27]. Indeed, when niclosamide was subjected to the *Salmonella typhimurium* microsomal test system, it was found to induce frameshift mutations in TA98 and TA1538 strains [28,29,30]. Similarly, many nitroaromatic derivatives such as nitrophenol, nitrobenzyl chloride, and nitroaniline are also weakly mutagenic in the Ames test or rec- assay [27,31]. For most of these compounds, their mutagenicity potential partly arises from their ability to form DNA adducts, causing DNA replication stress and DNA damage [27]. The nitroaromatic groups undergo bioactivation via nitroreduction to form various nitrogen-containing substituents [27]. This occurs largely in the GI tract, where nitroreductases are produced by gut microbiota [32,33]. The resultant nitrogen-containing compounds may then be absorbed by the GI tract and react with the DNA via nucleophilic substitution resulting in the formation of 8-hydroxydeoxyguanosine (8-OH-dG), or cause DNA damage such as cleavage at pyrimidines [29,33,34,35]. It remains to be determined if the aniline 4′-NO_2_ group on niclosamide may likewise trigger cellular DNA damage and confer nonspecific cytotoxicity in cells.

Several recent studies suggest that niclosamide inhibits cancer growth through multiple mechanisms, involving the modulation of Wnt/β-catenin, mTOR, STAT3, NF-kB, and Notch signalling pathways [34,35,36,37] as well as a targeted effect against p53-deficient cancers that is mediated through mitochondrial uncoupling and arachidonic acid-induced apoptosis [17]. While structure-activity studies using derivatives of niclosamide have revealed molecular features of niclosamide required for its antitumour mechanisms, mainly related to its protonophoric activity [13], it is unknown if niclosamide induces pleiotropic effects distinct from its protonophoric action. Evidence that aromatic nitro groups may impart nonspecific adverse toxicity raises concern that niclosamide may likewise trigger “off-target” toxicity and carcinogenicity. The lack of structure-activity study to examine properties of niclosamide and its consequence on DNA damage in cells prompted us to examine the cellular effects of niclosamide using several measurable parameters of genomic stability. Hitherto unreported, we describe the effects of niclosamide in the activation of genome-wide DNA damage and DNA breaks, precipitating in chromosomal instability. Remarkably, elimination of the aniline 4′-NO_2_ group reverses these phenotypes while preserving its targeted antitumour action, using p53-deficient cancer cells as a case in point. More importantly, our study suggests that the nonspecific and untargeted effects driven by the aniline 4′-NO_2_ group can be functionally uncoupled from its targeted mechanism of action mediated through mitochondrial uncoupling and concerted effects on calcium signalling, transcriptomics, and lipidomics changes. Altogether, our results demonstrate that collateral DNA damage accompanying niclosamide may be attenuated with ND-Nic and support the use of ND-Nic as a potentially safer alternative. Our results have broad implications for the use of niclosamide in cancers and other clinical applications.

## 2. Results

### 2.1. p53 Status Is a Determinant of Sensitivity to ND-Nic

Previously, we identified niclosamide from a library screen of FDA-approved compounds in their capacity to target p53-deficient cells, causing inhibition of p53-deficient cancer cells and tumour xenografts mediated through mitochondrial uncoupling [17]. While structure-activity experiments revealed that its protonophoric activity and mitochondrial uncoupling is a direct function of the hydroxyl group in the salicylic acid motif (Figure 1A), it is poorly understood if niclosamide may impose any pleiotropic effects unrelated to its protonophoric activity. In particular, we were concerned if the aniline 4′-NO_2_ group may confer any nonspecific cytotoxcity that is not directed through its previously described function in mitochondrial uncoupling, which was responsible for the selective targeting of p53-deficient cancer cells. A related concern is whether these nontargeted effects may compromise its targeted antitumour action and impart unintended side effects in normal cells harboring wildtype p53. A significant consequence that may arise contributing to adverse side effects is DNA damage. Collateral DNA damage can lead to normal cell toxicity and genomic instability, consequently resulting in the impairment of cellular function, cell loss, DNA mutagenesis, and malignant cell transformation [38,39,40,41].

To determine the potential effects of aniline 4′-NO_2_ group on DNA damage and cytotoxicity, we employed a nitro-deficient analogue of niclosamide, herein referred to as ND-Nic (Figure 1A), and compared it to niclosamide throughout this study. Although the hydroxyl group is the main site of protonophoric activity in niclosamide due to its dissociable proton, it is unclear if the aromatic 4′-NO_2_ affects its protonophoric activity and, more importantly, its selective targeting of p53-deficient cells. Therefore, we first sought to determine if ND-Nic retains its effects in selectively targeting the p53-deficient cells using a pair of isogenic cell lines HCT116 p53^+/+^ and p53^−/−^ that were well validated in assessing drug responses and DNA damage sensitivities due to p53 functionality [42]. The HCT116 p53^−/−^ cell line was generated by homologous recombination that resulted in a nullizygous deletion of the p53 gene [42] resulting in the loss of the wildtype p53 protein (Figure 1B).

We previously reported that niclosamide selectively targets p53-deficiency in cancers that result from either inactivating point mutations in p53 or a complete loss of p53 protein. The specificity of niclosamide for p53 loss provides a means to target a broad spectrum of p53 mutations in the clinic, including a variety of missense and nonsense mutations. Here, we showed that similar to niclosamide, ND-Nic elicited increased inhibition of HCT116 p53^−/−^ colony growth compared to its effects on HCT116 p53^+/+^ cells (Figure 1C). Almost complete growth inhibition (>90%) of HCT116 p53^−/−^ cells was observed at 7.5 µM of ND-Nic, whereas limited inhibition of HCT116 p53^+/+^ cells (<20%) was observed at the same drug concentration. The increased sensitivity of HCT116 p53^−/−^ cells to ND-Nic was also reflected in the bright field images of cells treated with the indicated drug concentrations at 48 h (Figure 1D). Similar results were seen when the effects of ND-Nic were studied in terms of its ability to induce apoptosis and cell death by Western Blot using an antibody specific for PARP1. During caspase-dependent apoptosis, full length PARP1, 116 kDa is cleaved into 89 kDa and 24 kDa fragments by caspases 3 and 7. Thus, the detection of the cleaved form of PARP1 at 89 kDa serves as one of the key indicators of apoptosis and is frequently used to detect the induction of an apoptotic form of cell death. Here, we demonstrated that treatment with ND-Nic (7.5 μM) results in an obvious induction of PARP1 cleavage in HCT116 p53^−/−^ cells (Figure 1E), and also a minimal extent of PARP1 cleavage was observed in HCT116 p53^+/+^ cells (Figure 1E). Densitometry was used to quantify the bands and ratios of intensities of cleaved PARP1 over full-length PARP1 which are shown at the bottom of the Western Blot. Further, detection of Annexin-V positivity, which marks early and late apoptotic cells, in ND-Nic treated cells revealed a dose-dependent increase in Annexin V positivity and to a greater extent in HCT116 p53^−/−^ cells, corroborating with the observation on PARP1 cleavage (Figure 1F). Together, our data strengthen the evidence that p53 is a critical determinant of drug response to niclosamide and its analogue ND-Nic.

It appeared that the removal of aniline 4′-NO_2_ group resulted in overall increased IC_50_ values in both p53^+/+^ and p53^−/−^ cells compared to niclosamide (Nic 1.6 and 0.9 μM, respectively, and ND-Nic 16.7 and 8.3 μM, respectively) (Figure 1H), suggesting perhaps that more ND-Nic may be necessary to achieve a similar dose effect compared to niclosamide. However, it remains to be determined if the apparently increased potency of niclosamide is achieved with a trade-off in its selectivity in targeting p53^−/−^ cells. To quantitatively determine the extent to which the 4′-NO_2_ moiety may interfere with its specific antitumour activity, we compared the drug responses of HCT116 wildtype and p53 knockout cells to ND-Nic and niclosamide using quantitive measurements of drug sensitivity.

A differential drug sensitivity score (dDSS) is calculated by the difference in areas under the drug response curves, which is outlined as the definite integral taken between two points on the *x*-axis, *a* and *b*, $\int_{a}^{b}f\left(x\right)dx.\;,$ where *f*(*x*) models the drug response at a given concentration x (Figure 1G) (refer to Materials and Methods). The integral dose-response over the defined dose range (for example, between a and b) is calculated as a continuous function of the multiple parameters of the nonlinear dose-response curve f(x) [43]. The quantitative scoring of differential drug sensitivity between patient and control cells has been applied to optimise the selection of cancer-selective drugs as well as drug-sensitive patient sub-groups. This has been shown to outperform other response parameters [43].

To enable the measurements of dDSS, we quantified the antiproliferative effects of niclosamide and ND-Nic using WST-1 cell proliferation assay. Cell viability was measured following treatment with increasing doses of niclosamide or ND-Nic in HCT116 p53^+/+^ and p53^−/−^ cells and percentage inhibition of cell growth was determined with DMSO-treated control cells taken as 100% (Figure 1H). Areas under the individual dose-response curves and dDSS between HCT116 p53^+/+^ and p53^−/−^ cells were calculated for each drug and shown in Figure 1H (refer to Materials and Methods). Whereas both niclosamide and ND-Nic promoted the selective inhibition of p53^−/−^ cells, ND-Nic resulted in a larger dDSS against p53-deficient cells (dDSS = 25.22) compared to niclosamide (dDSS = 2.43), implying that the therapeutic window for targeting p53^−/−^ cells is wider with ND-Nic. The benefits in augmenting the therapeutic window in order to achieve improved selectivity for cancer over normal cells is clearly important in the design of chemotherapeutic drugs and is a critical factor to be considered in both target selection and optimisation in drug design [44,45]. Using quantitative dDSS measurements, our results suggest that ND-Nic has improved selectivity against p53^−/−^ cells compared to niclosamide. Whether the enhanced target selectivity for p53^−/−^ cells might be a consequence of reduction or elimination of the nonspecific effects driven by the aniline 4′-NO_2_ group to wildtype p53 cells warrants further investigation. Consequently, this may have major implications for the selectivity of niclosamide for cancer cells over normal cells. Altogether, our data support ND-Nic as a promising structural analogue that enhances the selective targeting of p53-deficient cells through its mitochondrial uncoupling function. Our findings support the investigation into potential nonspecific cytotoxicity that may accompany the inclusion of the aniline 4′-NO_2_ group on niclosamide.

### 2.2. Niclosamide Induces Genome-Wide DNA Damage in Cancer Cells That Is Attenuated by the Elimination of Its Aniline 4′-NO_2_ Group

Next, we sought to determine if ND-Nic retained the mitochondrial uncoupling function. Mitochondrial uncoupling is a characteristic of niclosamide, which is also the key mechanism driving synthetic lethality in p53-deficient cells. Here, the use of a lipophilic cationic dye JC-1 allowed us to monitor the mitochondrial potential gradient (ΔΨm) in living cells. JC-1 monomers (green fluorescence) enter the mitochondria in living cells and accumulate forming JC-1 aggregates (red fluorescence) (Figure 2A). When the mitochondrial membrane potential is diminished or abrogated, either by mitochondrial uncoupling or in unhealthy apoptotic cells, the red/green fluorescence signal ratio is decreased. This phenomenon is observed in the presence of the classical oxidative phosphorylation uncoupler, carbonyl cyanide 3-chlorophenylhydrazone (CCCP) (Figure 2A). As expected, mitochondrial depolarisation induced by niclosamide led to a decrease in the red/green signal ratio (Figure 2B). Similarly, treatment with ND-Nic also significantly reduced the red/green signal ratio when compared to DMSO-treated control cells, confirming that ND-Nic led to mitochondrial membrane depolarisation (Figure 2C). We further demonstrated the roles of ND-Nic as mitochondrial uncoupler by measuring mitochondrial respiration using Seahorse Mito-Stress assay. Similar to FCCP positive control, ND-Nic induced mitochondrial uncoupling as observed from the increase in the oxygen consumption rate (OCR) (Figure 2D). Maximal respiration rates are calculated and presented for both HCT116 p53^+/+^ and p53^−/−^ cells, which revealed that there is no significant difference between cell lines after FCCP or ND-Nic induced mitochondrial uncoupling (Figure 2E). A hallmark of mitochondrial uncoupling is the depletion in cellular ATP levels as a result of a block in ATP synthesis without affecting the respiratory chain and proton-dependent ATP synthase. A sharp drop in intracellular ATP concentration is observed following treatment with ND-Nic, similar to that seen with niclosamide in HCT116 p53^+/+^ and p53^−/−^ cells (Figure 2F). These results are consistent with the observed depolarisation of mitochondrial membrane. This indicates that despite the elimination of the aromatic nitro group in ND-Nic, its function in uncoupling mitochondrial oxidative phosphorylation is preserved, an observation that is also consistent with the conservation of its selective targeting of p53-deficient cancer cells.

Next, to determine if niclosamide imposes any DNA damage and genotoxicity on cells, we carried out experiments to probe for cellular DNA damage using phospho-H2AX (γH2AX) as a reliable surrogate marker for DNA damage. Phosphorylation of γH2AX is dependent on ATM, a PI3K kinase, and is synonymous with the activation of DNA damage-dependent pathways in response to cellular insults [46,47]. Using an antibody specific to γH2AX, we observed a significant dose-dependent increase in γH2AX signals detected at a single-cell level using immunofluorescence (Figure 3A). Similarly, the phosphorylation of RPA32 at serine 4 and 8 (pRPAS4S8) in response to DNA breaks and replication stress, serves as another marker of the activation of DNA damage response [48,49,50]. Immunofluorescence detection under niclosamide treatment of pRPAS4S8 mirrored the trend observed in γH2AX whereby pRPAS4S8 levels are elevated in a dose-dependent manner (Figure 3B). Total RPA level detected using antibody against RPA70 showed a mild increase in RPA70 staining in response to niclosamide (Figure 3C). The activation of genome-wide damage by niclosamide was verified by Western Blot for γH2AX in whole-cell lysates (Figure 3D). Together, our results indicate that niclosamide induces genome-wide DNA damage, which is previously unreported. Remarkably, compared to niclosamide, treatment with ND-Nic significantly reduced γH2AX and pRPAS4S8 staining, suggesting that the aniline 4′-NO_2_ group is a major contributor to niclosamide-induced DNA damage response in cells.

To further determine if niclosamide induces the formation of DNA lesions, we employed comet assay. Comet assay enables the measurement of DNA strand breaks at the cellular level based on the ability of broken DNA strands to migrate towards the anode under an electric field. This gives rise to characteristic comet tails which are formed by fragmented DNA migrating away from the nuclei and can be visualised using conventional DNA stains [51]. The extent of DNA damage is quantified by the olive tail moment (OTM). This parameter represents the product of the percentage of total DNA in the comet tail and the difference in the comet tail and head lengths [51]. Hence, a higher OTM indicates a greater extent of DNA damage and strand breaks. Using doxorubicin (1 μM) as a positive control, we demonstrated that treatment with this drug for 24 h increased the OTM value over the untreated and DMSO treated cells (Figure 3E). Congruent with the detection of DNA damage response markers in niclosamide-treated cells, we observed an increase in DNA fragmentation associated with increasing concentrations of niclosamide that was reflected in the OTM values (~27 to ~55) (Figure 3E). Remarkably, in comparison, removal of the aniline 4′-NO_2_ group in ND-Nic significantly attenuated the level of DNA fragmentation, reducing OTM values to the range of ~3 to ~9 (Figure 3E).

The reduction in genotoxicity of the ND-Nic derivative was further supported by the dramatic suppression of micronuclei (MN) formation. MN are recognised as small extra-nuclear bodies consisting of either damaged chromosome fragments from acentric chromatid/chromosome fragments or whole chromatids/chromosomes. The fragments from whole chromatids/chromosome trail behind replicated DNA during anaphase indicating chromosomal instability that results from mitotic errors or DNA damage [52]. Assessment of MN has been widely exploited in genotoxicity screening ranging from carbon nanotubes to cancer therapeutics including Adriamycin, Gemcitabine, and Topotecan [53,54,55] and are used as biomarkers of genotoxicity, cancer risk, and tumour grade [52,56,57,58,59,60]. Here, we demonstrated that niclosamide induces the formation of MN at all concentrations tested within 48 h of drug exposure (Figure 3F). In contrast, the extent of chromosomal instability and MN formation was significantly reduced with ND-Nic treatment (Figure 3F).

Altogether, these results are consistent with the notion that the aniline 4′-NO_2_ group in niclosamide potentiates DNA damage and genotoxicity, independently of its function in mitochondrial uncoupling. More importantly, elimination of the aromatic nitro group was sufficient to attenuate DNA damage, as demonstrated using multiple molecular parameters to measure genotoxicity. This suggests that the aniline 4′-NO_2_ group is a major contributor to niclosamide-induced cellular DNA damage.

### 2.3. ND-Nic Induces Calcium Fluxes That Potentiate Its Antitumour Mechanism

So far, our results suggest that the antitumour action of niclosamide against p53-deficient cancer cells known to be mediated through its phenolic hydroxyl group may be uncoupled from its induction of cellular DNA damage that is driven by its aniline 4′-NO_2_ group. To strengthen this conclusion, we next aimed to determine the potential mechanism that might underlie the synthetic lethal killing of p53-deficient cancer cells by ND-Nic. We previously identified a potential mechanism underlying the selective killing of p53-deficient cells by niclosamide. This is mediated through a newly characterised pathway involving concerted changes in calcium flux and phospholipid turnover that ultimately resulted in a lethal accumulation of arachidonic acid metabolites and PARP1/caspase-3 dependent apoptosis [17]. Mitochondrial uncoupling has been shown to disrupt calcium homeostasis, and modulate cellular calcium signalling in physiological and pathological conditions [61]. Changes in the mitochondrial membrane potential can affect the ability of mitochondria to take up Ca^2+^, thereby affecting mitochondrial-dependent sequestration of cytoplasmic Ca^2+^ and intracellular calcium concentrations [62]. Therefore, the uncoupling of mitochondrial oxidative phosphorylation can result in various metabolic changes that compromise cell viability. Studies of calcium levels and tumour cell metabolome post niclosamide treatment showed that these changes include the induction of calcium fluxes and the accumulation of arachidonic acid prior to cell death [17,63]. We therefore asked if the mechanism of action of niclosamide mediated through calcium pathways is retained in ND-Nic. To monitor changes in intracellular calcium levels, we used the Fluo-4, AM dye, which is a cell-permeable and fluorescent Ca^2+^ indicator. Entry of Fluo-4, AM into the cell is followed by cleaving of the acetoxymethyl (AM) ester to give the free fluorescent indicator, Fluo-4, which cannot exit the cell. On binding Ca^2+^, the fluorescence of Fluo-4 is greatly increased and is detectable by flow cytometry. Upon the addition of the niclosamide, an increase in intracellular Ca^2+^ is detected in both p53^+/+^ and p53^−/−^ cells compared to untreated control cells (Figure 4A). To explore if ND-Nic induces any change in intracellular Ca^2+^ levels, we exposed p53^+/+^ and p53^−/−^ cells to ND-Nic before loading cells with the Fluo-4, AM dye. ND-Nic induced an increase in intracellular calcium levels in a dose-dependent manner, and, similar to niclosamide, the increased intracellular Ca^2+^ levels were detected in both HCT116 p53^+/+^ and p53^−/−^ cells (Figure 4B). To determine if the induced calcium flux is related to the growth inhibitory effects of ND-Nic, we deployed carbacyclin, a prostacyclin analogue that has been shown to antagonise calcium release from intracellular stores thus suppressing cytosolic calcium levels [64]. Similar to niclosamide [17], here we demonstrated that co-treatment with carbacyclin counteracted ND-Nic-induced cell growth inhibition and partially restored the colony growth of p53-deficient HCT116 cells (Figure 4C).

### 2.4. Lipidomics Profiling Revealed That ND-Nic Induces Accumulation of Long-Chain Fatty Acids and Arachidonic Acid in p53-Deficient Cancer Cells

Thus far, the results are consistent with the suggestion that ND-Nic functionally phenocopies niclosamide in targeting p53-deficient cancer cells through its mitochondrial uncoupling activity. Niclosamide induces differential lipid accumulation in p53-deficient cancer cells [18]. The selective inhibition of p53-deficient cancer cells is also shown to arise from an arachidonic acid-dependent mechanism [18,65,66]. To this end, we performed an untargeted lipidomics profiling of HCT116 p53^+/+^ and p53^−/−^ cells treated with ND-Nic and evaluated the drug-induced changes in lipid levels. Lipid species were extracted from ND-Nic treated or non-drug treated controls and subjected to tandem liquid chromatography-mass spectrometry (LC-MS) analysis. Differential analytes pre- and post-treatment with ND-Nic were identified and hierarchical clustering of lipid species for ND-Nic treated and control samples derived from HCT116 p53^+/+^ and p53^−/−^ cells was performed (Figure 5A). PCA analysis is widely used to visualise high-dimensional metabolomic data in 2D or 3D space. Principal component analysis (PCA) plots revealed drug-induced changes in the lipidomics profiles of HCT116 p53^+/+^ and p53^−/−^ cells (Figure 5B). It is shown that mitochondrial uncoupling results in global metabolome changes affecting various metabolic pathways in glycolysis, pentose phosphate pathway, lipid, and amino acid metabolism [67]. Similarly, the global changes in lipidomics profiles between ND-Nic treated cells and control cells (Figure 5A,B) are expected from the effects of ND-Nic on mitochondrial uncoupling.

Analysis of the lipid metabolite changes in HCT116 p53^+/+^ and p53^−/−^ cells revealed that long-chain fatty acids, lysophospholipids, and triglycerols were differentially enriched post-treatment with ND-Nic, which was also observed previously with niclosamide [17]. Of particular importance is the observed enrichment of arachidonic acid (20:4 (ω-6)) in p53^−/−^ cells (Figure 5C), which was previously found to promote the apoptosis of p53^−/−^ cells [17]. As with niclosamide, a pronounced enrichment of other fatty acids including docosatetraenoic acid (22:4 (ω-6)) and eicosapentaenoic (EPA) (20:5 (ω-3)), was also observed in p53^−/−^ cells. Of note, the presence of other long chain polyunsaturated fatty acids, including EPA, was also found to induce cellular apoptosis [68]. Other than de novo FA synthesis, the rapid emergence of these metabolites shortly after the addition of ND-Nic suggest that they may be liberated from intracellular storage, i.e., phospholipid stores. This is consistent with the observation that lysophospholipids (lysophosphatidylcholines (LysoPCs) and lysophosphatidylethanoamines (LysoPEs)) were also specifically enriched in HCT116 p53^−/−^ cells compared to p53^+/+^ cells post ND-Nic treatment, similar to that observed with niclosamide [17]. Lipidomics profiling also revealed lipid metabolites that are not affected by ND-Nic, including linoleic acid 18:2 and dodecanoic acid 12:0 (Figure 5D). Altogether, the results suggest that ND-Nic mimics the effects of niclosamide in inducing a steady-state accumulation of arachidonic acid and lysophospholipids that is promoted by p53 loss.

### 2.5. ND-Nic Activates the p53-Dependent Transcriptional Activity

To further delineate the mechanism of ND-Nic, we next determined its effects on p53 and its transcriptional activity. We previously reported that p53 plays an important role in arachidonic acid catabolism, thus accounting for the effects of loss in p53 function to the elevation in arachidonic acid levels in response to niclosamide. Accordingly, niclosamide induces the transactivation of p53-dependent target genes, including lipoxygenase genes, *ALOX5* and *ALOX12B*, which promote the turnover of arachidonic acid and thus prevent a lethal accumulation of arachidonic acid in cells with a functional p53 pathway. High levels of arachidonic acid have been demonstrated to cause mitochondrial membrane permeability transition and caspase 3-dependent apoptosis [63]. To examine if ND-Nic promotes the transactivation of p53-dependent target genes, we first examined the effects of ND-Nic on the canonical p53-target genes *p21* and *PUMA*. As shown in Figure 6A, there was a clear induction of *p21* and *PUMA* gene transcripts in HCT116 p53^+/+^ cells after treatment with ND-Nic, as was observed with niclosamide, suggesting that p53 is transcriptionally activated in response to ND-Nic. Indeed, their dependency on p53 is reflected by the pronounced difference in *p21* and *PUMA* gene transcript levels between HCT116 p53^+/+^ and p53^−/−^ cells. Similarly, we detected a significant induction of *ALOX5* and *ALOX12B* gene transcripts in response to ND-Nic and a greater extent in p53^+/+^ compared to p53^−/−^ cells (Figure 6B). This suggests that the induction of *ALOX5* and *ALOX12B* genes is dependent on p53, a notion that is consistent with the previously reported observation in niclosamide studies and also in Figure 5B [17]. The results suggest that p53 target genes, *ALOX5* and *ALOX12B*, were similarly induced by ND-Nic. Further work will be needed to determine if the transcriptional regulation of *ALOX5* and *ALOX12B* is critical for a p53-dependent response to ND-Nic, as was previously demonstrated for niclosamide [17]. Altogether, our results suggest that ND-Nic functionally phenocopies the cellular effects of niclosamide and parallels the action of niclosamide in eliminating p53-deficient cancer cells through a calcium-dependent pathway.

## 3. Discussion

Several studies have explored the structure-activity relationship of niclosamide derivatives to understand the contributions of various functional groups on niclosamide to its antitumour activity [14,69,70]. Here, we functionally dissected and characterised the effects of the aniline 4′-NO_2_ group on niclosamide’s cellular activity. Most studies have focused on elucidating structural changes that either enhance or compromise the effects of niclosamide on tumour cell killing and correlated that to its protonophoric activity. However, we found that niclosamide induces pleiotropic effects on DNA damage which is directed through its aniline 4′-NO_2_ group. It is clear that the antitumour mechanism of niclosamide is driven by its phenolic hydroxyl group and mitochondrial uncoupling, whereas the nonspecific genotoxicity is imposed by its nitro group on the aniline motif. The separation of the structure-function relationship therefore suggests that the two outcomes may be functionally uncoupled. In this study, we investigated the effects of niclosamide on DNA damage and genotoxicity of cells, using several molecular parameters and readouts of genomic and chromosomal instability. Interestingly, we found that the aniline 4′-NO_2_ group contributes significantly to DNA damage induced by niclosamide, which is independent of niclosamide’s function in mitochondrial uncoupling and in driving apoptosis in p53-deficient cancer cells. Removal of the aniline 4′-NO_2_ group diminished DNA breaks and chromosomal damage induced by niclosamide while enhancing its selectivity against p53-deficient cancer cells, using isogenic human colorectal HCT116 p53^+/+^ and p53^−/−^ cancer cells as a study model. To our knowledge, this is the first study focused on the structure-function analysis of niclosamide and its effects on DNA damage. The data presented here support the suggestion that the nonspecific genotoxicity driven by the aniline 4′-NO_2_ group can be functionally uncoupled from its protonophoric activity responsible for the targeted inhibition of p53-deficient cancer cells. This has further implications for future drug optimisation efforts to increase target specificity and selectivity. Apart, the result also raises the possibility that the apoptotic outcomes elicited by niclosamide may be a composite effect of various cell inhibitory activities operating through independent mechanisms and in part driven by genotoxic stress. This has added implications for potential side effects on normal cells that may influence the selectivity of niclosamide for cancer over normal cells.

Niclosamide is a protonophore that acts by translocating protons across the mitochondrial membrane, thus dissipating the mitochondrial membrane potential gradient and uncoupling oxidative phosphorylation from ATP synthesis [65,71,72]. Its mild mitochondrial uncoupling activity is shown to be tolerable in organisms [22,65] and is required for mechanisms promoting tumour cell inhibition, attenuating obesity, and mounting an antiviral response [73,74,75]. SAR studies have been performed to investigate the various substituents on niclosamide and their influence on its specific biological activities in cells [66]. Of key importance is the presence of the dissociable phenolic hydroxyl group and an amide proton, which forms an intramolecular hydrogen bond to stabilise the anionic form of niclosamide [13]. During respiration, protons are pumped out of the mitochondrial matrix into the intermembrane space. In its ionic form, niclosamide can associate with protons in the intermembrane space and transport them across the outer mitochondrial membrane, releasing them into the cytoplasm. This dissipates the proton gradient across the IMM. As a result, oxidative phosphorylation is uncoupled. We and others have demonstrated that the removal of the phenolic hydroxyl group or its replacement with a methoxy group (-O-CH_3_) completely abrogates its mitochondrial uncoupling activity measured in the Seahorse Mito-stress test assay. This informs the functional status of mitochondria by coupling the readout on cellular bioenergetics with mitochondrial inhibitors [25]. Studies have shown that the protonophoric activity of niclosamide was directly correlated to its effects in cytoplasmic acidification and the inhibition of mTOR signalling [13] which could have direct consequences on its antitumour activity. Apart from that, we previously demonstrated that the protonophoric activity of niclosamide is critical for the targeted inhibition of p53-deficient cancer cells, driven by a calcium signalling and arachidonic acid (Ca^2+^/AA) pathway [17]. Other protonophores, including FCCP, also selectively target p53-deficient cells through a Ca^2+^/AA pathway, supporting the view that selectivity against p53-deficient cells is a direct function of protonophore-induced lipidomics changes.

Here, in light of our finding that niclosamide induces DNA damage and chromosomal instability due to its aniline 4′-NO_2_ group, we re-examined the structural features of niclosamide required to elicit tumour cell apoptosis. We exploited our earlier finding that niclosamide exhibits increased selectivity in targeting p53-deficient cells, which was mediated through concerted effects on calcium levels and lipidomics changes in the Ca2^+^/AA pathway. Using quantitative measurements of differential drug sensitivity scores (dDSS) in human colorectal HCT116 p53^+/+^ and p53^−/−^ cancer cells, we compared the differential drug response. dDSS has been used to quantify the selective response of cancer cells, relative to that of control cells to provide an objective comparison between drugs and select those that are more cancer specific. Such scoring analysis parallels the use of the activity area metric to measure drug sensitivity [43,76] and outperforms other commonly used parameters such as IC_50_. In this study, we used dDSS scoring to provide an unbiased comparison between niclosamide and ND-Nic on their effects in targeting p53-deficient cells. Interestingly, the results revealed that ND-Nic exhibits improved drug selectivity for p53-deficient cells (dDSS = 25.22) compared to niclosamide (dDSS = 2.43), which is also reflected in the increased difference in areas under their drug dose-response curves (AUC) with ND-Nic (Figure 1H). The dDSS analysis suggests that ND-Nic may be more target selective against p53-deficient cells, but whether the enhanced selectivity against p53-deficient cells is a result of minimising unspecific genotoxicity to p53 wildtype cells remains to be determined. The empirical IC_50_ values would however imply that ND-Nic may be a less potent drug (IC_50_^(ND-Nic)^ p53^+/+^ = 16.7 μM; p53^−/−^ = 8.3 μM compared to IC_50_^(Nic)^ p53^+/+^ = 1.6 μM; p53^−/−^ = 0.9 μM). These interpretations may be reconciled by the earlier suggestion that the strong electron-withdrawing nature of the nitro-group may modulate the mitochondrial uncoupling potential of niclosamide [13]. The phenol group on niclosamide can become deprotonated in the slightly alkaline environment of the mitochondrial matrix. Stabilisation of the phenolate by a hydrogen bond to the amide group (-NH) would allow niclosamide to cross the IMM. In the intermembrane space of mitochondria, the lower pH environment will then allow the protonation of the phenolate and niclosamide could cross the inner membrane and deposit the proton in the mitochondrial matrix. The repeated process of this exchange of protons leads to the dissipation of the proton gradient. It was suggested that the electron-poor aniline ring may help to stabilise this process by modulating the pKa of the amide proton (-NH). Our data is congruous with this suggestion and the previous finding that the removal of the nitro group in niclosamide diminished its ability to interfere with mTORC1 signalling when compared at the same drug dose. This suggests potentially that the aniline 4′-NO_2_ group may modulate the protonophoric activity of niclosamide. Nevertheless, the enhanced selectivity of ND-Nic for p53-deficient cells adds another dimension for consideration and proposes that the apparent reduction in potency of ND-Nic is compensated by a gain in its specific targeting of p53-deficient cancer cells. These results have further implications for the role of genotoxicity in compromising the targeted effects of niclosamide that may be extrapolated to its selectivity for cancer over normal cells.

We had earlier demonstrated that niclosamide induces mitochondrial uncoupling, and through concerted effects on hydrolysis of phospholipids mediated by calcium-dependent phospholipases [17]. The induction of apoptosis in p53-deficient cells is eventually precipitated by a lethal accumulation of arachidonic acid due to a deficiency in the transcription of *ALOX5* and *ALOX12B*. To further substantiate our findings that ND-Nic retains its antitumour activity against p53-deficient cells, we demonstrated that ND-Nic functionally phenocopies niclosamide. This is reflected in its uncoupling of mitochondria, induction of intracellular calcium flux, and changes in lipidomics and transcriptomics, and is consistent with the mechanism of niclosamide in inducing selective apoptosis of p53-deficient cancer cells mediated through the Ca^2+^/AA pathway [17]. Since a large majority of cancers present with mutation of *Tp53*, our findings have significant implications for cancer therapeutics in the clinical setting [77]. In particular, in cancers that are often characterised by p53 mutations such as non-small-cell lung cancer, type II ovarian cancer, and a subset of acute myeloid leukaemia with exceedingly poor prognosis [69,70,78].

## 4. Materials and Methods

### 4.1. Cell Culture and Reagents

Human colon carcinoma cell lines, HCT116 p53^+/+^ and p53^−/−^ cells were kindly provided by Dr Bert Vogelstein (John Hopkins University School of Medicine, Baltimore, MD, USA). Derivative lines were routinely cultured in McCoy’s 5A modified media (Cytiva, USA SH30200.01), supplemented with 1% Penicillin-Streptomycin (Thermo Fisher Scientific, USA 15140) and 10% foetal bovine serum (FBS) (Hyclone, USA SV30160.03) incubated at 37 °C and 5% CO_2_.

### 4.2. WST-1 Cell Proliferation Assay

Cell viability was measured using WST-1 Cell Proliferation Assay (Abcam, USA ab65473). A total of 1500 cells were seeded onto a 96-well plate and were seated for 24 h before drug treatment with different concentrations of niclosamide (Sigma-Aldrich, USA N3510) and nitro-deficient niclosamide: 5-chloro-N-(2-chlorophenyl)-2-hydroxy-benzamide (Selena Chem, Ukraine SEL10044108) for 48 h. Cells were subsequently recovered in McCoy’s 5A media for 7 d. WST-1 reagents were prepared as recommended by the manufacturer’s protocol. Then, 10 µL of WST-1 reagent was added to each well and cells were incubated with WST-1 reagent at 37 °C and 5% CO_2_ for 1 h. Absorbance readings were measured at 440 nm using BioTek 800 TS absorbance reader.

### 4.3. Colony Forming Assay

A total of 50,000 cells per well were seeded onto 6-well plates. Cells were seated for 24 h before 48 h treatment of niclosamide at 0, 1, 2, 3, 4, and 6 µM and ND-Nic at 0, 2.5, 5, 7.5, 10, and 15 µM. Single or co-treatment of 6, 12, and 24 µM carbacyclin was carried out in cells treated with or without 7.5 µM ND-Nic for 48 h. Cells were recovered for 7 d in fresh media and washed with 1× phosphate-buffered saline (PBS) before staining with 1× crystal violet (Sigma-Aldrich USA C6158) for 2 h at room temperature. Cells were rinsed with distilled water after removal of crystal violet solution before image acquisition.

### 4.4. ATP Level Measurement

A total of 150,000 cells per well were seeded onto a 6-well plate and left to sit for 24 h. Cells were treated with 2 µM niclosamide and 5 µM ND-Nic for 48 h. Cells were then harvested via trypsinisation and lysed in nucleotide releasing buffer (Abcam, USA ab65313). ATP levels were obtained using ADP/ATP Ratio Bioluminescent Assay Kit (Abcam, USA ab65313). Luminescence signal was measured using BioTek Cytation3 Imaging Reader.

### 4.5. Fluo-4, AM Calcium Measurement

A total of 150,000 cells per well were seeded in 6-well plates. Cells were treated with 2 µM niclosamide and 5, 7.5, and 10 µM of ND-Nic for 16 h. According to the manufacturer’s protocol, cells were harvested, pelleted, then resuspended in 500 µL of 2 μM Fluo-4, AM (Thermo Fisher Scientific, USA F14201) media solution and incubated for 30 min at 37 °C and 5% CO_2_. Cells were subsequently centrifuged and resuspended in 1× PBS before proceeding to flow cytometry analysis.

### 4.6. Western Blot

A total of 1,000,000 cells were seeded in a 10 cm dish. Then, 2 and 4 µM niclosamide or 7.5 and 15 μM ND-Nic was added to the cells for 48 h post 24 h of cell seeding. Proteins were extracted via lysis in ice-cold RIPA buffer (Thermo Fisher Scientific, USA 89901), supplemented with 1× proteinase cocktail inhibitor (Roche, USA 05892791001), 1 mM DTT (Bio Basic Asia Pacific, Singapore DB0058), 1 mM sodium fluoride (Merck Millipore, USA 106450), 100 µM sodium orthovanadate (Sigma-Aldrich USA S6508), and 100 µM phenylmethylsulfonyl fluoride (Sigma-Aldrich USA P7626). Cells were sonicated for 20 s at low amplitude followed by centrifugation. The supernatant was collected for the determination of protein concentration using the Pierce BCA protein assay kit (Thermo Fisher Scientific USA 23227). Extracted proteins were separated by 15% SDS-PAGE and transferred onto a PVDF membrane. The membrane was incubated with the following primary antibodies overnight at 4 °C: mouse monoclonal anti-Hsp90 (BD transduction, USA; 610418; 1:5000), mouse monoclonal anti-p53 (Santa Cruz Biotechnology, USA; DO-1 sc-126; 1:2500), mouse monoclonal anti-γH2AX (Merck Millipore; 05-636; 1:1000), mouse polyclonal anti-pRPAS4S8 (Bethyl Laboratories USA; A300-245A; 1:500), rabbit monoclonal anti RPA70 (Abcam USA; ab79398; 1:2000) and rabbit polyclonal anti PARP (Cell Signalling USA; 9542S; 1:1000). PVDF membrane was subsequently washed for 10 min in 1× PBS with 0.1% Tween-20 (Promega USA H5152) 3 times prior to incubation in horseradish peroxidase-conjugated secondary antibodies (Agilent Dako USA P0161, P0448; 1:5000) for 2 h at room temperature. ECL detection reagent (Cytiva Amersham USA 45-00-999) prepared as recommended by the manufacturer’s protocol was used for the visualisation of specific protein expression on X-ray film after 3 times of 10 min washes in 1× PBS with 0.1% Tween.

### 4.7. Quantitative Real-Time PCR

A total of 500,000 cells were seeded onto a 5 cm dish and left to sit for 24 h. Either 2 and 4 µM niclosamide or 5, 7.5, and 15 µM ND-Nic were added to cells for 48 h. Cells were lysed in TRIzol reagent (Invitrogen USA 15-596-018) with RNA extracted using phenol:chloroform:isoamyl alcohol (Merck Millipore USA P3803) and RNeasy mini kit (Qiagen USA 74136). cDNA was generated using 2 µg of total RNA and reverse transcriptase using a two-step RT-PCR protocol. The cDNA was amplified by qPCR using SSO Advanced Universal Green SYBR mix (Bio-Rad USA 1725274) and specific primers in a thermal cycler with the following set up: 95 °C for 3 min followed by 40 cycles of 95 °C for 30 s, 60 °C for 30 s and 72 °C for 30 s. β-actin was used as the housekeeping gene. The specific primer sequences are listed as Table 1.

### 4.8. Immunofluorescence

A total of 100,000 cells per well were seeded on a coverslip (22 × 22 mm) in a 6-well plate. 48 h of treatment in either 2, 4, and 6 µM niclosamide or 5, 7.5, and 15 µM nitro-deficient niclosamide were performed after 24 h of cell seeding. The cells were washed with 1× PBS before fixation in 4% paraformaldehyde (*w*/*v*) (Sigma Aldrich USA 47608) for 10 min. The cells were washed 3 times with 1× PBS before permeabilisation with 0.5% Triton X-100 (Promega USA H5141) for 10 min at room temperature. The coverslips were washed in 1× PBS for 5 min 3 times before blocking with 3% bovine serum albumin for 1 h. Cells were washed in 1× PBS for 5 min 3 times and stained with primary antibody: mouse monoclonal anti-γH2AX (Merck Millipore USA; 05-636; 1:700) overnight for 4 °C. After that, the coverslips were washed 3 times for 5 min each with 1× PBS followed by secondary antibody: anti-mouse (Alexa Fluor 488; Thermo Fisher Scientific USA; A11029) for 2 h in the dark. Nuclei were stained with DAPI (Invitrogen USA 33342) for 5 min before mounting onto poly-lysine slides (Thermo Fisher Scientific USA J2800AMNZ). Imaging and quantification analysis were done using Zeiss AxioImager.Z1 and Image J, respectively. MN were identified using DAPI stained nuclei to access the genotoxicity of niclosamide and nitro-deficient niclosamide. Identified MN were carefully scored based on the morphology, size, and their proximity to the nucleus. MN frequency is evaluated as the number of MN counted/total number of cells × 100.

### 4.9. Comet Assay

A total of 100,000 cells were seeded onto a 6-well plate and treated for 24 h in 1 µM doxorubicin and 48 h in either 2, 4, and 6 µM niclosamide or 5, 7.5, and 15 µM nitro-deficient niclosamide after 24 h of cell seeding. Comet Assay (Cell Biolab, Inc USA STA-351) was performed according to the manufacturer’s protocol. Lysis buffer (1× lysis solution (provided), 100 mM EDTA, 2.4 M NaCl, pH 10), alkaline solution (300 mM NaOH, 1 mM EDTA) was prepared and chilled before use. SuperFrost Gold glass slides (Thermo Fisher Scientific USA K5800AMNZ72) were pre-coated with 1% low melting agarose (Promega USA PR-V2111) and warmed at 37 °C before use. Cells were harvested via scraping with ice-cold 1× PBS. Harvested cells were counted and resuspended to obtain a final concentration of 1.27 × 10^6^ cells/mL before diluting with comet agarose in 1:10. 75 µL of the cell-agarose mixture was added to the glass slides and left to solidify for 15 min at 4 °C. The slides were immersed into the pre-chilled lysis buffer for 90 min at 4 °C followed by incubation in alkaline solution for 30 min at 4 °C. Prior to the electrophoresis, the slides were immersed in pre-chilled 1× TBE solution twice for 5 min each. The TBE electrophoresis was carried out at 24 V for 20 min at 4 °C. The slides were washed twice for 2 min in pre-chilled deionised water and 70% ethanol for 5 min before air drying. Once the slides were dried, 100 µL of Vista green DNA dye (diluted 1: 10,000 TE buffer) was added directly onto the gel spot and incubated for 15 min in the dark. Imaging was performed using Zeiss AxioImager.Z1 fluorescent microscope with FITC filter. Tail length and head length were measured to calculate the olive tail moment (OTM) according to the following equation: (tail length-head length) x% tail DNA. The average OTM for each sample was obtained through analysis and measurement of 100 cells per condition.

### 4.10. Mitochondrial Uncoupling Assay

A total of 100,000 cells were seeded onto a 6-well plate and left seated for 24 h. Cells were treated with 50 µM carbonyl cyanide 3-chlorophenylhydrazone (CCCP), 2 and 4 µM niclosamide and 7.5 and 15 µM ND-Nic for 3 h. As recommended by the manufacturer’s protocol, 2 µM JC-1 dye (Life Technology USA M34152) was added directly to all samples and incubated for 15 min at 37 °C and 5% CO_2_. Live cell imaging was performed using Zeiss LSM800 inverted confocal microscope with heating chamber to capture staining of JC-1 at 488 nm (green) and J aggregates at 595 nm (red).

### 4.11. Drug Sensitivity Score Analysis

A percentage of cell inhibition was extrapolated from dose-response data obtained from WST-1 assay and converted into R programming language to obtain dose-response curves and parameters required for the generation of the area under the curve (AUC). The formula of AUC was derived from the integration of the concentration when the drug-response curve crosses at least 10% of the activity level, ‘*t*’ following Bhagwan Yadav et al.’s 4-parameter logistic function as shown below [43].
$$\mathrm{AUC}=a\left[x_{2\;}-c+\;\frac{log_{10}\left(1+10^{b\left(c-x_{2}\right)}\right)+log_{10}\left(1-\;\frac{t}{a}\right)}{b}\right],$$
where ‘*a*’ stands for the maximal response, R max, ‘*b*’ stands for the slope of the % cell inhibition curve, ‘*c*’ stands for the half-maximal inhibitory concentration, IC_50,_ ‘*d*’ stands for the minimal response that is the bottom asymptote of the % cell inhibition curve, and ‘$x_{2\;}$’ stands for maximum concentration. Differential AUC (*dAUC*, the difference in AUC between two dose-response curves) was then used for the calculation of dDSS, which describes the difference in drug sensitivity between two cell lines in this context.


$$\mathrm{dDSS}\;=\frac{dAUC-t\left(x_{2}-x_{1}\right)}{\left(100-t\right)\left(C_{max}-C_{min}\right)}.$$


Therefore, differential drug sensitivity score was then calculated based on the above formula of AUC normalisation with ‘$x_{2}$’ and ‘$x_{1}$’ standing for the maximum and minimum concentration that shows a differential response while ‘*C_max_*’ and ‘*C_min_*’ stand for the maximum and minimum concentration tested.

### 4.12. Lipidomics Profiling

The protocol is taken with reference to Kumar et al. [17]. In brief, 2 × 10^7^ cells per sample were collected and gently washed with ice-cold 150 mM sodium chloride (Merck, Darmstadt Germany) to quench cellular metabolism. Lipid species were extracted using a two-phase liquid-extraction protocol as previously described [73]. Briefly, methanol, chloroform, and 3.8 mM Tricine (Merck USA 39468) solution (approx.1:1:0.5 *v*/*v*) was used to separate polar metabolites (aqueous) from lipid species (organic fraction). Lipid metabolites in the organic fraction were stored in 2 mL amber glass vials and the headspace was filled with nitrogen gas to minimise sample degradation. All extracts were stored at −80 °C and subsequently analysed using a LC-MS approach as described [73]. Raw LC-MS data obtained were pre-processed using the XCMS, an abbreviation for various forms (X) of chromatography mass spectrometry, peak finding algorithm. Total area normalisation was applied to the pre-processed data prior to statistical analysis using multivariate (SIMCA-P+ software) and univariate tools, including relative ratios, Student’s *t*-test (Welch′s correction), and hierarchical clustering for the classification of common trends.

### 4.13. Mitochondrial Cellular Respiration

A total of 40,000 cells were seeded per well on Agilent Seahorse XF96 cell culture microplate (Agilent USA 101085-004). Sensor Cartridge was hydrated the overnight before measuring mitochondrial cellular respiration. Cells were recovered in Agilent Seahorse XF Base medium (pH 7.4) (Agilent USA 103334-100) supplemented with 20 mM glucose (Sigma USA G8270) and 2 mM pyruvate (Sigma USA S8636). Oxygen consumption rates were measured by CF Analyzer (Searhorse Bioscience) at 37 °C with the injection of 2 µM oligomycin, 0.5 µM FCCP or 7.5 µM ND-Nic, and 0.5 µM antimycin A into the ports of the XF assay cartridge (Agilent 103729-100).

### 4.14. Annexin-V Apoptosis Assay

A total of 100,000 cells were seeded per well in 6-well plates and left seated for 24 h. Cells were treated with ND-Nic for 48 h, respectively. According to the manufacturer’s protocol, cells were harvested, pelleted, and resuspended in 130 µL of freshly prepared Annexin-V-Fluos labelling solution comprising Annexin-V-Alexa Fluor 568 (Roche Diagnostics USA 03 703 126 001) and HEPES incubation buffer (Roche Diagnostics USA 11 858 777 001) at a 1:50 (*v*/*v*) ratio. After 20 min of incubation in the dark at room temperature, 300 µL of 1× PBS was added to each sample before proceeding to flow cytometry analysis (BD Biosciences FACSymphony™ A5 Cell Analyzer) using a 610/30 BP filter.

## 5. Conclusions

Altogether, the study herein presents evidence that nitro-deficient analogue of niclosamide retains the antitumour mechanism of niclosamide while minimising off-target effects due to DNA damage. Collateral DNA damage and genotoxicity imposed on normal cells by chemotherapy often limits clinical efficacy [39]. Therefore, the option for uncoupling nonspecific genotoxicity from its targeted mechanism of action by ND-Nic presents a potential strategy for evading general toxicity to normal tissues. This improves clinical efficacy and warrants further investigation into similar analogues for future drug development.

## Figures and Tables

**Figure 1 ijms-22-10420-f001:**
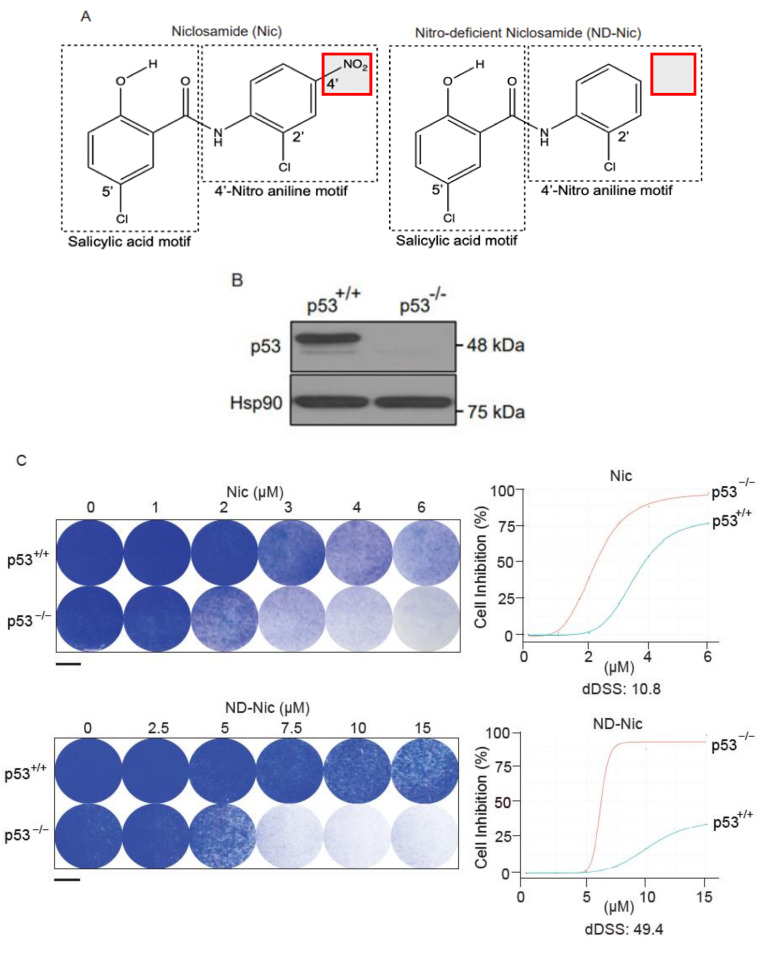
Nitro-deficient niclosamide retains the selective killing capacity against p53-deficient cells. (**A**) Chemical structures of niclosamide (Nic), 5-chloro-N-(2-chloro-4-nitrophenyl)-2-hydroxybenzamide, and nitro-deficient niclosamide (ND-Nic), 5-chloro-N-(2-chlorophenyl)-2-hydroxy-benzamide. The red box highlights the nitro group on the aniline motif that is eliminated in the ND-Nic analog. (**B**) p53 protein expression in HCT116 p53^+/+^ and p53^−/−^ cells detected by Western Blot. Hsp90 was used as the loading control. (**C**) Crystal violet staining compares the survival of human colon carcinoma HCT116 p53^+/+^ and p53^−/−^ cells 7 d post recovery from 48 h treatment with Nic and ND-Nic. The graph quantifies the colony intensity of the crystal violet staining results. Black scale bar at 37 mm. (**D**) Bright field microscopic images of HCT116 p53^+/+^ and p53^−/−^ cells treated with ND-Nic are shown at indicated concentrations after 48 h. Black scale bar at 100 μm. (**E**) Protein expression of apoptosis marker, cleaved PARP1 was detected by Western Blot in HCT116 p53^+/+^ and p53^−/−^ cells treated with either DMSO, 2 μM Nic or 7.5 μM ND-Nic for 48 h. GAPDH was used as the loading control. Ratios of intensities of cleaved PARP1 over full-length PARP1 are shown quantified at the bottom. (**F**) Annexin-V-Alexa Fluor 568 is used to detect the early and late apoptotic cells induced in response to ND-Nic in HCT116 p53^+/+^ and p53^−/−^ cells using flow cytometry. HCT116 p53^+/+^ and p53^−/−^ cells treated with 0, 5, 7.5, 10, and 12.5 μM ND-Nic for 48 h. Error bars represent ± SD of at least 3 independent experiments. *t*-test was used to determine significance (****, *p* < 0.0001; ns, *p* > 0.05). (**G**) Schematics of the low and high differential drug sensitive score (DSS) plotted from cell inhibition (%) against drug concentration (μM) (refer to Materials and Methods). dDSS denotes the differential drug response of two cell lines tested in comparison. (**H**) Cell inhibition (%) was assessed in HCT116 cells treated with Nic and ND-Nic at the indicated concentrations for 48 h using WST-1 cell proliferation assay. Raw data were converted into R programming language and plotted to generate area under the curve (AUC) for the calculation of differential AUC (dAUC) and differential drug sensitivity score (dDSS). Approximate IC_50_ values are IC_50_^(Nic)^ p53^+/+^ = 1.6 μM; p53^−/−^ = 0.9 μM) and IC_50_^(ND-Nic)^ p53^+/+^ = 16.7 μM; p53^−/−^ = 8.3 μM). Error bars represent ± SD of at least 3 independent experiments.

**Figure 2 ijms-22-10420-f002:**
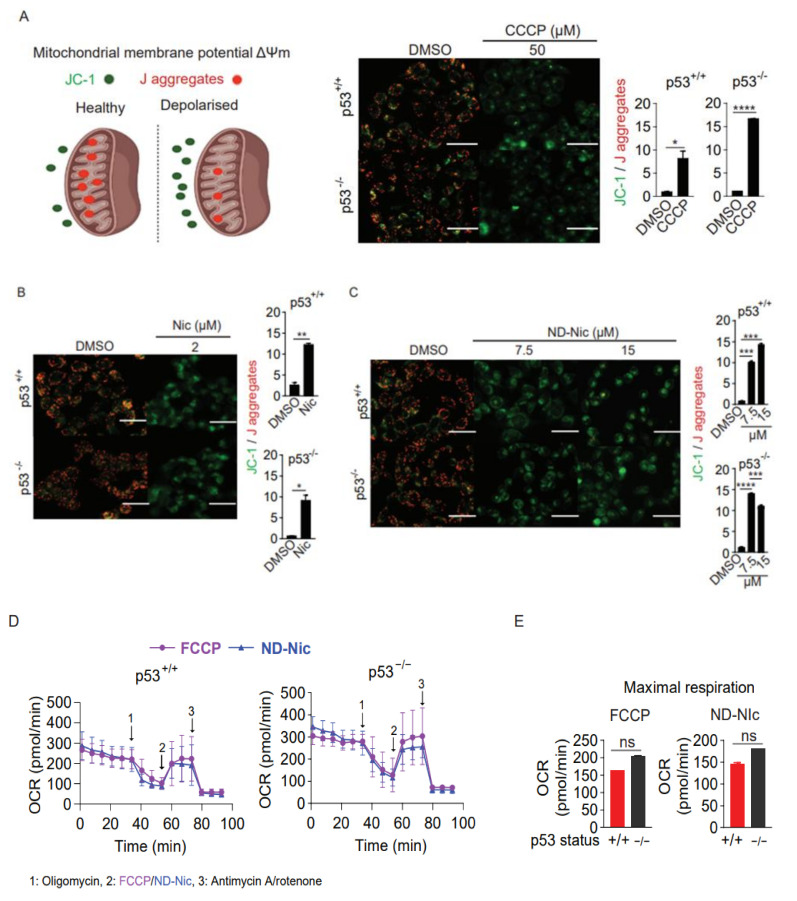
Nitro-deficient niclosamide analogue shares similar mitochondrial uncoupling activity with niclosamide (**A**) Schematics of mitochondrial membrane potential determined by the mean intensity of JC-1 (green) over J aggregates (red). Healthy mitochondrial membrane potential enables the inward transport of JC-1 to form J aggregates while depolarised mitochondrial membrane results in the accumulation of JC-1 outside of the mitochondria with little to no J aggregates formation. Representative immunofluorescence images and quantification of JC-1 to J aggregates ratio in human colon carcinoma HCT116 cells treated with either 0.4% DMSO or 50 µM carbonyl cyanide 3-chlorophenylhydrazone (CCCP) for 3 h. White scale bar at 50 μm. Representative immunofluorescence images and quantification of JC-1 to J aggregates ratio in HCT116 cells treated with either (**B**) 0.4% DMSO and 2 µM Nic or (**C**) 0.4% DMSO and 7.5 and 15 µM ND-Nic for 3 h. White scale bar at 50 μm. Error bars represent ± SD of at least 3 independent experiments. *t*-test was used to determine significance (*, *p* < 0.05; **, *p* < 0.01; ***, *p* < 0.001; ****, *p* < 0.0001). (**D**) Oxygen consumption rate (OCR) measured in HCT116 p53^+/+^ and HCT116 p53^−/−^ cells using Seahorse Mito Stress Assay by adding ATP synthase inhibitor, oligomycin A and mitochondrial uncouplers, 0.5 µM FCCP or 7.5 µM ND-Nic. (**E**) Bar graph of maximal respiration of HCT116 p53^+/+^ and HCT116 p53^−/−^ induced by the addition of 0.5 µM FCCP and 7.5 µM ND-Nic. Error bars represent the ± SD of at least 3 independent experiments. *t*-test was used to determine significance (ns, *p* > 0.05). (**F**) Percentage of ATP production in 0.4% DMSO, 2 μM Nic and 5 μM ND-Nic treated HCT116 cells for 48 h. Error bars represent ± SD of at least 3 independent experiments. *t*-test was used to determine significance (****, *p* < 0.0001).

**Figure 3 ijms-22-10420-f003:**
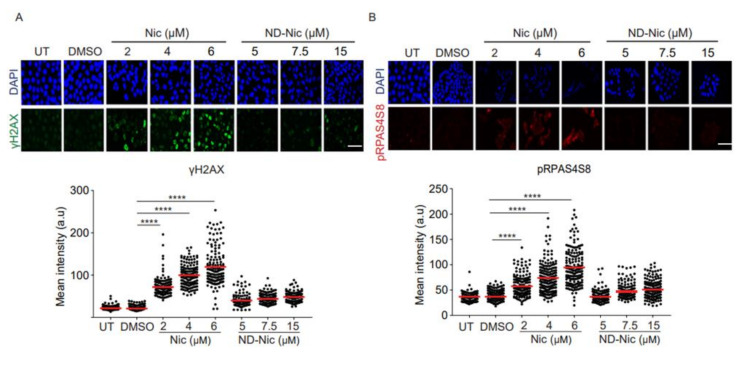
Elimination of the aniline 4′-NO_2_ group attenuated genome-wide DNA damage induced by niclosamide. Representative immunofluorescence images and quantification of DNA damage markers, (**A**) γH2AX, (**B**) pRPAS4S8, (**C**) and RPA70 in human colon carcinoma HCT116 cells, untreated (UT) or treated with 0.4% DMSO, Nic (2, 4, and 6 µM) and ND-Nic (5, 7.5, and 15 µM) for 48 h. White scale bar at 50 μm. Error bars represent ± SD of at least 3 independent experiments. *t*-test was used to determine significance (****, *p* < 0.0001). (**D**) Western Blots of γH2AX, p53, and Hsp90 (loading control) in HCT116 cells, untreated (UT) or treated with Nic (2 and 4 µM) and ND-Nic (7.5 and 15 μM) for 48 h. (**E**) Left: Representative fluorescent images of Vista green stained DNA obtained from comet assay in untreated (UT), positive control doxorubicin (Doxo), 0.4% DMSO, Nic, and ND-Nic treated HCT116 cells. Bottom: A magnified image of comet tail emerging from a nucleus. Tail length is indicated in red. Right: Olive tail moment (OTM), calculated as (tail length-head length) x% tail DNA quantifies the extent of breaks detected by the comet assay in HCT116 cells, untreated (UT) or treated with 1 µM Doxo (24 h), DMSO, Nic (2, 4, and 6 µM) and ND-Nic (5, 7.5, and 15 µM) for 48 h. White scale bar at 50 μm. Error bars represent ± SD of at least 3 independent experiments. *t*-test was used to determine significance (** = *p* < 0.01 and **** = *p* < 0.0001). Average olive tail moment values are: UT (2), 1 µM Doxo (58), DMSO (3), Nic 2 µM (27), 4 µM (46), and 6 µM (55), ND-Nic 5 µM (3.5), 7.5 µM (7), and 15 µM (9). (**F**) Representative images of DAPI stained nuclei in untreated (UT), 0.4% DMSO, Nic, and ND-Nic treated HCT116 cells for 48 h. White arrow indicates micronuclei. Bottom left: White arrow indicates a micronucleus formed adjacent to a parent nucleus. Bottom right: Percentage of micronuclei detected in HCT116 cells, untreated (UT) or treated with DMSO, Nic (2, 4, and 6 µM) and ND-Nic (7.5 and 15 µM) for 48 h. White scale bar at 50 μm. Error bars represent ± SD of at least 3 independent experiments. *t*-test was used to determine significance (* = *p* < 0.05 and ** = *p* < 0.01).

**Figure 4 ijms-22-10420-f004:**
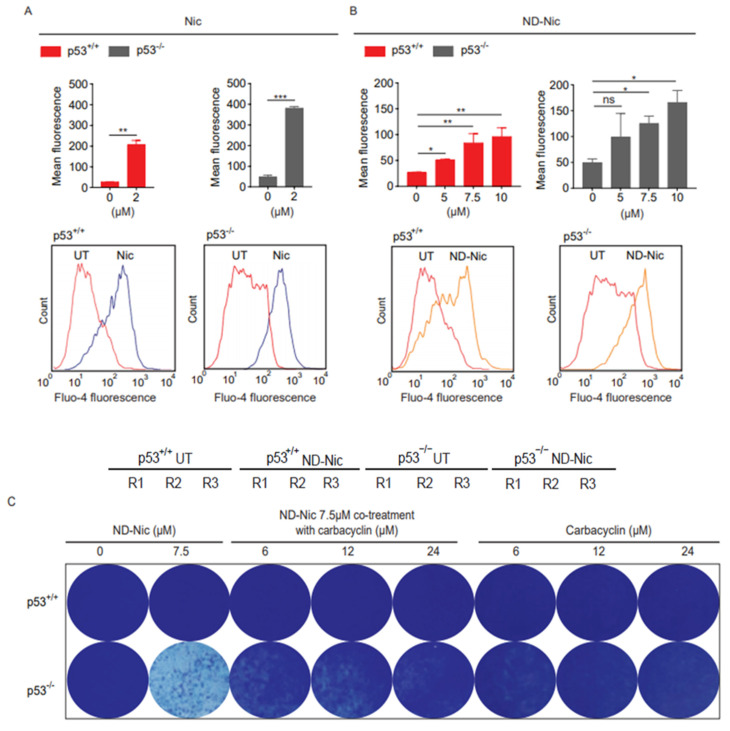
Nitro-deficient niclosamide induces calcium flux independent of p53 status. Calcium flux analysis using Fluo-4, AM was conducted in human colon carcinoma HCT116 p53^+/+^ and p53^−/−^ cells, untreated or treated with (**A**) 2 μM Nic and (**B**) ND-Nic (5, 7.5, and 10 μM) for 16 h. Mean fluorescence intensities (Top) were quantified using FlowJo software. (Bottom) FlowJo software plots of calcium flux detected by the shift of Fluo-4 fluorescence in either (**A**) 2 μM Nic or (**B**) 10 μM ND-Nic treated HCT116 p53^+/+^ and p53^−/−^ cells. Error bars represent ± SD of at least 3 independent experiments. *t*-test was used to determine significance (*, *p* < 0.05; **, *p* < 0.01; ***, *p* < 0.001; ns, *p* > 0.05). (**C**) Crystal violet staining of HCT116 p53^+/+^ (Top) and p53^−/−^ (Bottom) 7 d post recovery from either single treatment with ND-Nic (7.5 μM) or ND-Nic 7.5 μM co-treatment with carbarcyclin (6, 12, and 24 μM) for 48 h. Treatment with carbacyclin alone serves as the negative control. Black scale bar at 37 mm.

**Figure 5 ijms-22-10420-f005:**
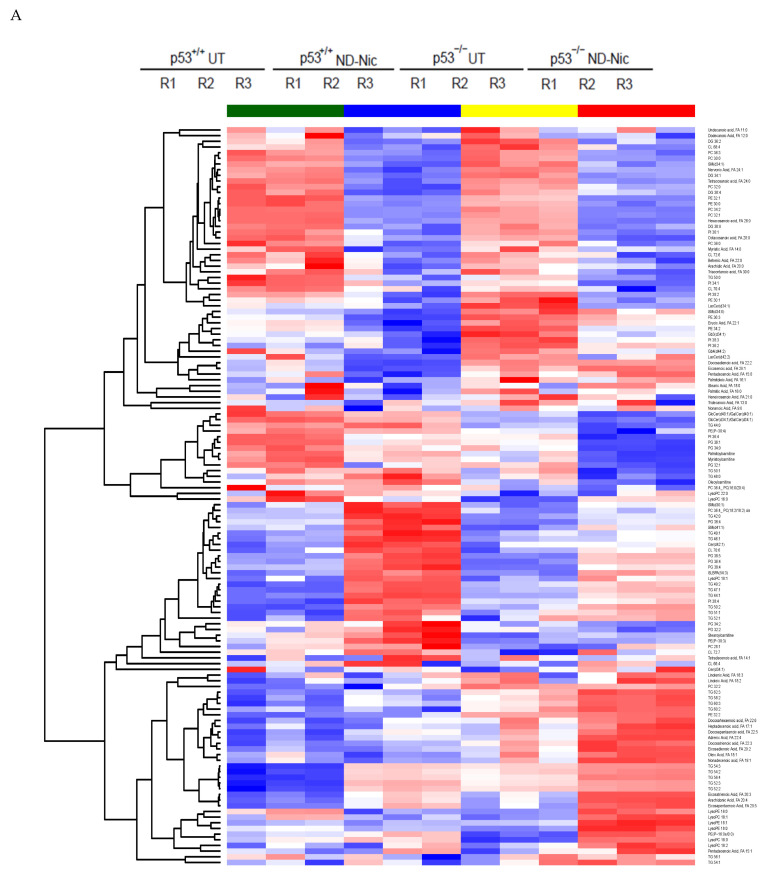
Nitro-deficient niclosamide induces increased lipid metabolite level in p53-deficient cells. (**A**) Hierarchical clustering representation of differential lipid metabolite levels between non-drug treated (UT) and 7.5 μM ND-Nic treated human colon carcinoma HCT116 p53^+/+^ and p53^−/−^ cells for 16 h. (**B**) Lipid metabolic composition was compared between 7.5 μM ND-Nic (16 r) denoted as (treated) and untreated (untreated) HCT116 p53^+/+^ and p53^−/−^ cells using principal component analysis scores plot. (**C**) Bar graphs display enriched lipid metabolite levels in 7.5 μM ND-Nic treated HCT116 p53^−/−^ cells for 16 h measured by LC-MS including fatty acids (arachidonic acid 20:4, adrenic acid 22:4, eicosapentaenoic acid 20:5, eicosatrienoic acid 20:3, and eicosadienoic acid 20:2) and lysophospholipids (lysophosphatidylethanoamines (LysoPEs): LysoPE 20:3, LysoPE 22:2, and LysoPE 18:1 and lysophoaphatidylcholines (LysoPCs) LysoPC 14:0, LysoPC 26:1, and LysoPC 16:1). Error bars represent ± SD of at least 3 independent experiments. *t*-test was used to determine significance (*, *p* < 0.05; **, *p* < 0.01; ***, *p* < 0.001; ****, *p* < 0.0001; ns, *p* > 0.05). (**D**) Examples of other lipid metabolites unaffected by ND-Nic treatment are shown. Bar graphs display levels of linoleic acid 18:2 and dodecanoic acid 12:0 in 7.5 μM ND-Nic treated HCT116 p53^−/−^ cells for 16 h. Error bars represent ± SD of at least 3 independent experiments. *t*-test was used to determine significance (ns, *p* > 0.05).

**Figure 6 ijms-22-10420-f006:**
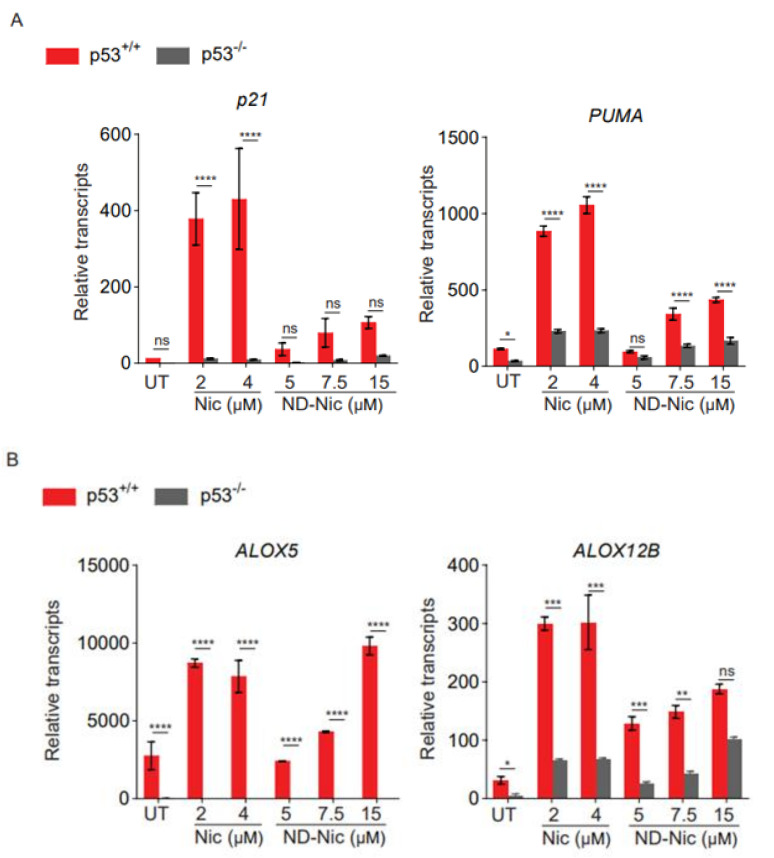
Nitro-deficient niclosamide induces p53 dependent upregulation of *Alox5* and *Alox12B*. Relative transcript level of (**A**) *p21* and *PUMA* and (**B**) *Alox5* and *Alox12B* assessed by qPCR in human colon carcinoma HCT116 p53^+/+^ and p53^−/−^ cells, untreated (UT) or treated with Nic (2 and 4 μM) and ND-Nic (5, 7.5, and 15 μM) for 48 h. Error bars represent ± SD of at least 3 independent experiments. *t*-test was used to determine significance (*, *p* < 0.05; **, *p* < 0.01; ***, *p* < 0.001; ****, *p* < 0.0001; ns, *p* > 0.05).

**Table 1 ijms-22-10420-t001:** The specific primer sequences.

*Gene*	qPCR Primers
*Alox5*	Forward: 5′–GGTACCTGAAGTACATCACGCTGA–3′
	Reverse: 5′–CGTCGGTGTTGCTTGAGAATGTGA–3′
*Alox12B*	Forward: 5′–ACGGCCGTATCTACCACTTC–3′
	Reverse: 5′–AGTCCTGCTTGGCTCTGATC–3′
*p21*	Forward: 5′–GAAAACGGCGGCAGACCAGC–3′
	Reverse: 5′–TGTAGAGCGGGCCTTTGAGG–3′
*PUMA*	Forward: 5′–ACGACCTCAACGCACAGTACG–3′
	Reverse: 5′–TCCCATGATGAGATTGTACAGGAC–3′

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
