# Peer review of "Nitro-Deficient Niclosamide Confers Reduced Genotoxicity and Retains Mitochondrial Uncoupling Activity for Cancer Therapy"

_ijms, 2021, doi:10.3390/ijms221910420_

Round 1

Reviewer 1 Report

Nitro-deficient niclosamide confers reduced genotoxicity and retains mitochondrial uncoupling activity for cancer therapy

In the study, the authors analyzed the cellular activities of aniline 4’-NO2 group of niclosamide. Elimination of the nitro group in ND-Nic analogue reduced γH2AX signals and DNA breaks while preserving its antitumour mechanism mediated through a calcium signalling pathway and arachidonic acid metabolism. Elimination of its nitro group enhanced the target selectivity of niclosamide against p53 deficiency. The findings are interesting, but some drawbacks as following could be improved.

  1. The treatment concentrations of Nic and ND-Nic are different and are not consistent in all experiments. It is not clear why these doses are used in different experiments. The authors should clearly mention. The authors might rearrange the Results section, and show the data of Figure 3 first in the Result section before the other data. The readers will understand the different effects of the two agents at different concentrations.
  2. In Figure 5, it is not clear for the concentration of ND-Nic treatment. It should be shown in the Figure legend.
  3. The Figure 6 panel affects the figure legend.
  4. For the HCT116 p53-/- cell line, the authors should clearly mention whether its p53 is null (p53 gene deletion) or just loss of p53 transcription activity. One concern is the some effects of Nic and ND-Nic on the expression of PUMA and ALOX12B in the p53-/- cells in Figure 6.
  5. For real-time PCR in Figure 6, it is not clear for the large scale units of relative transcript in the y-axis.
  6. The protein expression and function of ALOX5 and ALOX12B in the lipid metabolites should be demonstrated.

Author Response

Dear Reviewer 1, 

     Thank you for your comments and suggestions which have substantially improved our manuscript. We have addressed the comments in the attached point-by-point response. Please see the attachment. 
     Thank you! 

Best Regards,
CCF lab  

Reviewer 2 Report

Tsz Wai Ngai and Gamal Ahmed Elfar et al. have investigated the role of the aniline 4’-NO2 group of the antihelminth drug niclosamide (5-chloro-N-(2-chloro-4-nitrophenyl)-2-hydroxybenzamide) (Nic) in modulating its antitumor activity. In comparison to Nic, nitro-deficient niclosamide (5-chloro-N-(2-chlorophenyl)-2-hydroxybenzamide) (ND-Nic), elicited decreased mitochondrial function in model human colon carcinoma HCT116 cells. Although ND-Nic showed decreased capacity to induce DNA damage compared to Nic, it displayed increased cytotoxicity toward p53-deficient HCT116 cells. This could be attributed to altered Ca2+ and lipid homeostasis. This finding potential opens a therapeutic window of ND-Nic as an anti-cancer agent with reduced genotoxicity toward healthy (p53-expressing) tissues. Despite the study is well designed, the data are robust, and text extremely well written and illustratively laid out, several key claims need to be further strengthened by experiments. Namely, whether ND-Nic is a bona fide mitochondrial uncoupler and apoptotic activator needs to be more specifically tested (see the major points below).

Major points:

1) The authors used untreated cells as controls in Figures 1B–E, 2–E, however more proper control would be vehicle-treated cells with vehicle representing the solvent used to dissolve the particular drug of interest and added at exactly the same volume and for the same period of time as with the drug treatment.

2) Despite the authors have demonstrated decreased mitochondrial aggregation of JC-1 following Nic and ND-Nic treatment in Figures 1C,D, it is not clear whether Nic and ND-Nic stimulate mitochondrial respiration? This data would better support the role of these compounds in inducing mitochondrial uncoupling. 

3) Although the authors have demonstrated increased immunofluorescent pRPAS4S8 staining following NIC treatment in Figure 2B, it is not clear whether this was indeed due to increased phosphorylation of RPA32 at serine 4 and 8, as they claim, or due to increased levels of the total pRPAS protein?

4) There seems to be problem with the quantitation of Fluo-4 fluorescence in Figures 4A and 4B since the AUC for ND-Nic in both p53+/+ and p53-/- cells is more or less similar to the AUC observed with 2 uM Nic in p53-/- cells in the flow cytometry plots, however this is displayed as ~100 and ~150 for ND-Nic in p53+/+ and p53-/- cells, respectively, and ~400 for 2 uM Nic in p53-/- cells in the quantitated graphs.

5) The evidence that ND-Nic induces apoptotic form of cell death in p53-/- cells is missing.

6) Please show western blot showing the p53 status of p53+/+ vs. p53-/- HCT116 cells as part of a new figure/panel.

Minor points:

1) Please replace "confers" with something like "induces" (line 19).

2) Please change "5-chloro-N-2-chloro-4-nitrophenyl)-2-hydroxybenzamide" to "5-chloro-N-(2-chloro-4-nitrophenyl)-2-hydroxybenzamide" (lines 32, 196).

3) "Recently, studies revealed that niclosamide elicits anti-neoplastic effects against cancer cell lines across broad cancer types including acute myelogenous leukaemia (AML) stem cells, lung, brain, ovarian, breast, and colorectal cancer [10-17] and in patient-derived xenografts and organoids [18, 19], unravelling the potential of re-purposing niclosamide as an antitumour agent, an attractive possibility given that niclosamide is FDA-approved with an established safety and pharmacokinetics profile" (line 38) is too long. Please split into two sentences.

4) Please define abbreviation for "FDA" (line 43), "LysoPE" (line 622), "LysoPC" (line 623), "AA" (line 738), "DI" (line 906), "Env" (line 920), "XCMS" (line 952), "CCF" (line 980), "IMCB" (line 980).

5) Please replace "gastrointestinal" with "gastrointestinal (GI)" (line 47) and "gastrointestinal (GI)" with "GI" (line 79).

6) "Whereas the aniline 4’-NO2 group may undergo nitroreduction yielding anilines (aromatic amines) that may be potentially mutagenic" (line 69).

7) Please change "p53-defective" to "53-deficient" (lines 101, 337, 409, 487, 489, 491, 615).

8) Please replace "mitochondria" with "mitochondrial" (lines 139, 635).

9) The resolution of micrographs in Figures 1B–D; 2A,B,D,E; 3B is poor. Please replot each image panel using higher pixel density so that individual cellular features can be clearly distinguished. Alternatively, provide these high pixel density micrographs as part of new supplementary figures.

10) Please indicate statistical significance in Figures 1B,C,D,E; 2A,B,D,E; 4A,B; 5C; 6A,B.

11) Please replot the JC-1 quantitation panels in Figure 1B with identical y-axis scales.

12) Please replace "J-aggregates" with "J aggregates" (line 199).

13) It is not clear from the legend to Figures 1B, 1C, and 1D how long were HCT116 cells treated with 50 uM CCCP, 2 μM Nic, and 7.5 μM or 15 μM ND-Nic, respectively?

14) Similarly, it is not clear how long were HCT116 cells treated with 2 μM Nic and 5 μM ND-Nic in Figure 1E?

15) Please replace "is" with "was" (line 219).

16) It is not clear what the authors mean by "ND-Nic significantly reduced γH2AX and pRPAS4S8" in "Remarkably, compared to niclosamide, treatment with ND-Nic significantly reduced γH2AX and pRPAS4S8, suggesting the aniline 4’-NO2 group is a major contributor to niclosamide-induced DNA damage response in cells" (line 221)? Do they actually mean "ND-Nic significantly reduced γH2AX and pRPAS4S8 levels", "ND-Nic significantly reduced γH2AX and pRPAS4S8 staining", or "ND-Nic significantly reduced γH2AX and pRPAS4S8 expression"?

17) Please change "Using doxorubicin as a positive control, we demonstrated that treatment with doxorubicin (1 μM)" to something like "Using doxorubicin (1 μM) as a positive control, we demonstrated that treatment with this drug" (line 233).

18) Although the authors claim that OTM values were in the range ~30 to ~55 in "Congruent with the detection of DNA damage response markers in niclosamide-treated cells, we observed an increase in DNA fragmentation associated with increasing concentrations of niclosamide treatment that was reflected in the OTM values (~30 to ~55)" (line 235), the range is indicated as ~27 to ~55 in the legend to Figure 2D.

19) Please change "genotoxicity in" to "genotoxicity of" (line 241).

20) The vertical y-axis labels "DAPI", "γH2AX", "pRPAS4S8" are not exactly vertically centered to the accompanying image panels in Figures 2A,B,E. Please align more perfectly.

21) Please reverse the order of the western blot image panels shown in Figure 2C so that γH2AX is on top, p53 in the middle, and Hsp90 on the bottom of the plot.

22) The vertical y-axis labels "Hsp90", "p53" are not exactly vertically centered to the accompanying western blot image panels in Figure 2C. Please align more perfectly.

23) Please provide scale bars for the zoomed in images in Figures 2D,E.

24) Please replace "micronuclei detection" with "micronuclei" (line 313).

25) Please change "NO2" to "4’-NO2" (lines 325, 340).

26) Please replace "Drug Sensitivity Score" with "drug sensitivity score" (lines 343, 357, 418, 938).

27) Please change "M&M" to "Materials and Methods" (lines 346, 358, 414).

28) It is not clear what the authors mean by "WST1 assay" (line 353)?

29) Please replace "differential Drug Sensitivity Score (dDSS)" with "dDSS" (lines 357, 938).

30) Please change "Refer" to "refer" (line 358).

31) The sentence "Using quantitative dDSS measurements, our results suggest that ND-Nic has improved selectivity against p53-/- cells compared to niclosamide (dDSSND-Nic =25.22 and dDSSNic=2.43)" (line 365) is duplicit to "Whereas both niclosamide and ND-Nic promoted the selective inhibition of p53-/- cells, it is clear from the results that ND-Nic resulted in a larger dDSS against p53-deficient cells (dDSS=25.22) compared to niclosamide (dDSS=2.43), implying that the therapeutic window for targeting p53-/- cells is wider with ND-Nic" (line 358). Please remove the latter or use it as a concluding sentence but without repeating the dDSS values.

32) Please replace "minimising or eliminating" with "reduction or elimination of" (line 368).

33) "Although it appeared that the removal of aniline 4’-NO2 group resulted in overall increased IC50 values in both p53+/+ and p53-/- cells compared to niclosamide (Figure 3D), suggesting perhaps that more ND-Nic may be necessary to achieve a similar dose-effect compared to niclosamide, it remains to be determined if the apparent increased potency of niclosamide is achieved with a trade-off in its selectivity in targeting p53-/- cells" (line 370) is too long. Please split into at least two parts.

34) It is not clear what the authors mean by "the latter possibility" in "Consequently, the latter possibility will have major implications for the selectivity of niclosamide for cancer cells over normal cells" (line 375)?

35) Figure 3A seems to be too pixelated. Please provide panels with enhanced resolution so that individual cell colonies can be discerned. Also, please replace quantitation graphs using comparable pixel density to those used in Figure 3D. Moreover, it would be good to have dAUC and dDSS parameters included under these plots in analogy to Figure 3D.

36) Although the concept of dDDS is sufficiently explained in Figure 3C, it is not clear how dAUC is calculated and the difference between these two parameters? Would it please be possible to include similar schematics depicting low and high dAUC in analogy to the plots explaining the difference between low and high dDDS in Figure 3C?

37) The line thickness of y-axes in Figure 3C is not uniform along the whole axis length since there is a short white "dent" present in each of the two plots. Please replot using uniform line thickness for both y-axes.

38) Please replace "Low dDSS" and "High dDSS" with "dDSS" in Figure 3D.

39) Please change "Drug Sensitive Score (DSS)" to "drug sensitivity score (dDSS)" (line 414).

40) Please change "phospholipids" to "phospholipid" (line 424).

41) Please replace "p53+ cells" with "p53-/- cells" (lines 442, 444).

42) It is not clear what the authors mean by "and in response to niclosamide" in "To determine if the induced calcium flux is related to the growth inhibitory effects of ND-Nic, we employed the use of carbacyclin, a prostacyclin analogue that is shown to antagonise calcium release from intracellular stores thus suppressing cytosolic calcium levels [67] and in response to niclosamide" (line 447)?

43) Please change "employed the use of" to "deployed" or "used" (line 448).

44) Please replace "is" with "has been" (line 449).

45) Please change "restored the colony growth of p53-deficient HCT116 cells, at least in part" to "partially restored the colony growth of p53-deficient HCT116 cells" (line 452).

46) The y-axes of the quantitated graphs in Figures 4A,B are labeled as "Mean fluorescence", whereas the respective figure legend claims that "Mean fluorescence (Top) was quantified from the area under curve generated in flow cytometry (Bottom)" (line 480). Please distinguish "mean fluorescence" from "AUC" as these are different quantitation parameters and relabel y-axes accordingly.

47) The x-axes of the quantitated graphs in Figures 4A,B are labeled as "Alexa Fluor 488", whereas the respective figure legend and the main text claims that Fluo-4 fluorescence was measured.

48) Please replot the Fluo-4 quantitation panels in Figures 4A,B with identical y-axis scales.

49) It is not clear from the flow cytometry curves in Figures 4A,B what concentration of Nic and ND-Nic, respectively, they represent?

50) Figure 4C is too pixelated. Please provide panels with enhanced resolution so that individual cell colonies can be discerned.

51) Please change "ND-Nic 7.5uM cotreatment with Carbacyclin (uM)" to "ND-Nic 7.5 uM co-treatment with carbacyclin (uM)" in Figure 4C.

52) Please swap the order of rows in the Crystal violet staining of p53-/- and p53+/+ cells plotted in Figure 4C so that p53+/+ cells are plotted in the top row and p53-/- cells are plotted in the bottom row.

53) Please replace "Fluo4" with "Fluo-4" (line 478).

54) Please change "under" to "under the" (line 480).

55) Please replace "in" with "by" (line 481).

56) Please expand the "LCFAs" abbreviation in the title "2.4. Lipidomics profiling revealed that ND-Nic induces accumulation of LCFAs and arachidonic acid in p53-defective cancer cells" (line 486).

57) Please change "DMSO-controls" to "DMSO-treated controls" (lines 496, 616).

58) "Differential analytes pre- and post-treatment with ND-Nic were identified and hierarchical clustering of lipid species for ND-Nic treated and control samples derived from HCT116 p53+/+ and p53-/- cells was performed (Figure 5A) to reveal key species that are enriched in p53-/- cells" (line 497) is too long. Please split into two sentences.

59) It is not clear what key species are the authors referring to in "Differential analytes pre- and post-treatment with ND-Nic were identified and hierarchical clustering of lipid species for ND-Nic treated and control samples derived from HCT116 p53+/+ and p53-/- cells was performed (Figure 5A) to reveal key species that are enriched in p53-/- cells" (line 497)?

60) The statement "Principal component analysis (PCA) plots reflected similar global lipid changes in drug-treated HCT116 p53+/+ and p53-/- cells" (line 501) does not make sense since it is hard to conceive how can one draw similarities between Figure 5A and 5B given each of the plots shows a different type of data?

61) It is not clear how Figures 5A or 5B can be consistent with the uncoupling of oxidative phosphorylation induced by ND-Nic in "This was expected and consistent with the fact that ND-Nic, similar to Nic, induces the uncoupling of oxidative phosphorylation in both wildtype and p53-deficient cells, resulting in extensive changes to the global lipidome profiles (line 502)"?

62) The statement "Next, we compared lipid metabolites induced in response to ND-Nic (line 506)" comes rather as a surprise since the extent of lipid metabolites induced by ND-Nic has been already discussed in the preceding paragraph (Figures 5A, 5B).

63) Although the authors claim that "Of particular importance is the observed enrichment of arachidonic acid (20:4 (ω-6)) in p53-/- cells, which was found to promote the apoptosis of p53-/- cells" (line 509), the result that arachidonic acid induces apoptosis in p53-/- cells is nowhere to be seen. Please fix.

64) "Other than de novo FA synthesis, the rapid emergence of these metabolites shortly after the addition of ND-Nic suggest that they may be liberated from intracellular storage, i.e., phospholipid stores, a contention consistent with the observation that lysophospholipids (lysophosphatidylcholines (LysoPCs) and lysophosphatidyl-ethanoamines (LysoPEs)) were also specifically enriched in HCT116 p53-/- cells compared to p53+/+ cells post ND-Nic treatment, similar to that observed with niclosamide" (line 515) is too long. Please split into two sentences.

65) Please replace "lysophosphatidyl-ethanoamines" with "lysophosphatidylethanoamines" (line 519).

66) Please explain in the text how to interpret Figure 5B?

67) Figures 5A,C are too pixelated. Please provide panels with enhanced resolution so that all labels can be clearly read.

68) It is not clear what does the cyan trace indicate in the color map in Figure 5A?

69) It is not clear what does count mean in the color map in Figure 5A?

70) Please comment on what does branch length in the clustering shown in Figure 5A represent?

71) Please explain why there are three balls (dark yellow, orange, red, and green) in each of the cluster seen in the PCA plot and what do these represent in Figure 5B?

72) It is also not clear what do the individual grouped clusters (transparent yellow, orange, red, and green) represent in Figure 5B?

73) Would it please be possible to include one example of a lipid that did not significantly change upon ND-Nic treatment in p53-/- cells as part of Figure 5C?

74) It is not clear using what concentration of ND-Nic and how long were cells treated for in Figures 5A,B,C?

75) Please replace "metabolites" with "metabolite levels" (line 615).

76) Please change "Treated" to "treated" (line 618).

77) Please replace "Untreated" with "untreated" (line 619).

78) Please change "Arachidonic" to "arachidonic" (line 621).

79) Please replace "Adrenic" with "adrenic" (line 621).

80) Please change "Eicosapentaenoic" to "eicosapentaenoic" (line 621).

81) Please replace "Eicosatrienoic" with "eicosatrienoic" (line 622).

82) Please change "Eicosadienoic" to "eicosadienoic" (line 622).

83) "metabolism" could be replaced with "catabolism" (line 629).

84) Please change "examine" to "examined" (line 637).

85) "Similarly, we detected a significant induction of ALOX5 and ALOX12B gene transcripts in response to ND-Nic and a greater extent in p53+/+ compared to p53-/- cells (Figure 6B), suggesting that the induction of ALOX5 and ALOX12B genes is dependent on p53, a notion that is consistent with the previously observed in niclosamide studies and also in Figure 5B" (line 642) is not grammatically correct with respect to "a notion that is consistent with the previously observed in". Please revise.

86) Please format "and" without italics (line 647).

87) Please replace "parallel" with "parallels" (line 651).

88) The vertical y-axis label "Relative transcripts" is not exactly vertically centered to the accompanying graph panels in Figures 6A,B. Please align more perfectly.

89) The legend to Figure 6 is obscured by two of its panels (line 679).

90) It is not clear what reference gene was used to compare the analyzed transcripts to in Figure 6?

91) It is not clear why ALOX5 transcripts are not seen in p53-/- cells in Figure 6B?

92) "The data presented here support the suggestion that the nonspecific genotoxicity driven by the aniline 4’-NO2 group can be functionally uncoupled from its protonophoric activity responsible for the targeted inhibition of p53-deficient cancer cells, which has further implications for future drug optimisation efforts to increase target specificity and selectivity" (line 705) is too long. Please split into two sentences.

93) Please change "mitochondria" to "inner mitochondrial" (line 716).

94) "Of key importance is the presence of the dissociable phenolic hydroxyl group and an amide proton which forms an intramolecular hydrogen bond to stabilise its anionic form" (line 722) is not semantically perfect since one would expect that "its" refers to "amide proton", while it is the anionic form of the dissociable phenolic hydroxyl group that is stabilized by hydrogen bonding.

95) The role of niclosamide in transferring protons across the outer mitochondrial membrane into the cytoplasm as suggested in "In its ionic form, niclosamide can associate with protons in the intermembrane space and transport them across the outer mitochondrial membrane, releasing them into the cytoplasm, hence disrupting the proton gradient across the inner mitochondrial membrane" (line 725) does not make any sense. Do the authors actually mean that niclosamide equilibrates protons across the inner mitochondrial membrane?

96) "We and others have demonstrated that the removal of the phenolic hydroxyl group or its replacement with a methoxy group (-O-CH3) completely abrogates its mitochondrial uncoupling activity measured in the Seahorse Mito-stress Test assay which informs of the functional status of mitochondria by coupling the readout on cellular bioenergetics with mitochondrial poisons" (line 729) is too long. Please split into two sentences.

97) Please replace "Test" with "test" (line 731).

98) The term "mitochondrial poisons" seem not to be appropriate (line 733).

99) Please use a different abbreviation to distinguish "arachidonic acid" (line 738) and activity area (line 752).

100) Please change "targets" to "target" (line 739).

101) Please replace "lipidomics changes" with "lipidomic changes" (lines 741, 746).

102) "p53+/+ and p53-/- cancer cells" is mentioned twice in "Using quantitative measurements of drug sensitivity scores (DSS) in the isogenic human colorectal HCT116 p53+/+ and p53-/- cancer cells, we compared the differential drug response of p53+/+ and p53-/- cancer cells" (line 747).

103) "Please change "drug sensitivity scores (DSS)" to "differential drug sensitivity scores (dDSS)" (line 747) and "Differential DSS (dDSS)" (line 749) to "dDSS" (line 749).

104) Please change "cancer-specific" to "cancer specific" (line 752).

105) Please replace "Activity Area" with "activity area" (line 752).

106) Please change "dDSS=25.22" to "dDSS = 25.22" (line 758).

107) Please replace "dDSS=2.43" with "dDSS = 2.43" (line 758).

108) It is not clear from "These interpretations may be reconciled by the earlier suggestion that the strong electron-withdrawing nature of the nitro-group may cause the dissociable proton of the phenolic group to be less acidic thus modulating the mitochondrial uncoupling potential of niclosamide" (line 764) how does more dissociable proton of the phenolic group in Nic or ND-Nic modulate uncoupling potential?

109) "Our data is congruous with this suggestion and the previous finding that the removal of the nitro group in niclosamide diminished its ability to interfere with mTORC1 signalling when compared at the same drug dose, suggesting potentially that the aniline 4’-NO2 group may modulate the protonophoric activity of niclosamide" (line 768) is too long. Please split into two sentences.

110) Please shorten "of the compound" to "of" (line 773).

111) Please provide reference for "We had earlier demonstrated that niclosamide induces mitochondrial uncoupling, and through concerted effects on hydrolysis of phospholipids mediated by calcium-dependent phospholipases" (Line 778).

112) "To further substantiate our findings that ND-Nic retains its antitumour activity against p53-deficient cells, we demonstrated that ND-Nic functionally phenocopies niclosamide which is reflected in its uncoupling of mitochondria, induction of intracellular calcium flux, and changes in lipidomics and transcriptomics, and is consistent with the mechanism of niclosamide in inducing selective apoptosis of p53-deficient cancer cells mediated through the Ca2+/AA pathway" (line 782) is too long. Please split into at least two sentences.

113) "Since a large majority of cancers present with mutation of Tp53, our findings have significant implications for cancer therapeutics in the clinical setting [80], especially in cancers that are often characterised by p53 mutations such as non-small-cell lung cancer, Type II ovarian cancer, and a subset of acute myeloid leukaemia with exceedingly poor prognosis" (line 788) is too long. Please split into two sentences.

114) Please change "Type" to "type" (line 791).

115) Please use standard spacing size (lines 812–819, 856–869, 871–889, 891–913, 915–921, 923–928, 957–964).

116) Please replace "Crystal" with "crystal" (lines 817, 819).

117) Please specify "nucleotide releasing buffer" (line 823).

118) It is not clear at what absorbance wavelengths was the luminescence signal read in "Luminescence signal was measured using BioTek 800 TS absorbance reader" (line 825)?

119) Please specify the catalog number of the protease inhibitor cocktail used in Proteins were extracted via lysis in ice-cold RIPA buffer (Thermo Fisher Scientific 89901), supplemented with 1x proteinase cocktail inhibitor, 1 mM DTT, 1 mM sodium fluoride, 100 μM sodium orthovanadate and 100 μM phenylmethylsulfonyl fluoride" (line 838).

120) Please specify the type of Triton used in "The cells were washed three times with 1x PBS before permeabilisation with 0.5% triton (Promega H5141) for 10 minutes at room temperature" (line 875).

121) Please replace "triton" with "Triton" (line 876).

122) Please replace "And" with "and" (line 885).

123) Please change "(the number of MN counted/total number of cells)" to "the number of MN counted/total number of cells" (line 888).

124) Please replace "Alkaline" with "alkaline" (lines 895, 903).

125) It is not clear what the authors mean by "It" in "It was counted and resuspended to obtain a final concentration of 1.27x106 cells/ml before diluting with comet agarose in 1:10. 75 μl of the cell-agarose mixture was added to the glass slides and was left to solidify for 15 minutes at 4°C" (line 899).

126) Please change "Lysis" to "lysis" (line 902).

127) "Volts" could be shortened to "V" (line 905).

128) Please replace "Dye" with "dye" (line 907).

129) From "The average olive tail moment for each sample was obtained through analysis and measurement of 50 comet tails" (line 911) is not clear how many cells were used for analysis?

130) Please change "100,000. cells" to "100,000 cells" (line 915).

131) Please replace "Inverted Confocal Microscope" with "inverted confocal microscope" (line 920).

132) Please change "Chamber" to "chamber" (line 921).

133) Please replace "488nm" with "488 nm" (line 921).

134) Please change "595nm" to "595 nm" (line 921).

135) Please format without italics (lines 943–955).

136) Please replace "tricine" with "Tricine" (line 947).

137) "Therefore, the option for uncoupling nonspecific genotoxicity from its targeted mechanism of action by ND-Nic presents a potential strategy for evading general toxicity to normal tissues to improve clinical efficacy and warrants further investigation into similar analogues for future drug development" (line 960) is too long. Please split into two sentences.

138) Please use initials for designating author contributions (lines 965–973).

139) Please include title and affiliation of Bert Vogelstein in the Acknowledgements section.

140) Please change "(Ming Zhou, Si Qi, Teo Jia Min, Foo Yan Ling, Ryan Wong Yong Ken and Leonard Goh XianYang) of the CCF lab/IMCB" to "of the CCF lab/IMCB: Ming Zhou, Si Qi, Teo Jia Min, Foo Yan Ling, Ryan Wong Yong Ken and Leonard Goh XianYang" or "of the CCF lab/IMCB, Ming Zhou, Si Qi, Teo Jia Min, Foo Yan Ling, Ryan Wong Yong Ken and Leonard Goh XianYang" (line 979).

Author Response

Dear Reviewer 2, 

     Thank you for your comments and suggestions which have substantially improved our manuscript. We have addressed the comments in the attached point-by-point response. Please see the attachment. 
     Thank you! 

Best Regards,
CCF lab  

Round 2

Reviewer 2 Report

Major points:

1) The PARP1 signal in the middle panel (DMSO p53 -/-, 7.5 uM ND-Nic p53 -/-) of Figure 1E is visibly cut through. Please either provide a vertical division bar to indicate that these are separate western blot lanes or completely replace with a new image.

2) Annexin-V assay lacks the mention of propidium iodide (PI), which also seems to be part of the Annexin-V-FLUOS Staining Kit. Please either remeasure and/or clearly indicate the definition of apoptotic cells with regards to the flow cytometry analysis. Early apoptotic cells should represent only the population that is Annexin-V-Alexa Fluor 568-positive and PI-negative.

Minor points:

1) Please change "single-dose" to "single dose" (line 52).

2) Please replace "inner mitochondrial membrane" with "inner mitochondrial membrane (IMM)" (line 63) and "mitochondrial membrane" (line 64) and "inner mitochondrial membrane" (lines 804, 845) with "IMM".

3) Please change "their mutagenicity" to "their" (line 75).

4) Please replace "abilities" with "ability" (line 76).

5) Please change "tract, and" to "tract and" (line 80).

6) Please replace "selectively targeting" with "their capacity to target" (line 113).

7) Please change "employed the use of" to "employed" or "deployed" (line 129).

8) Please replace "was" with "were" (line 135).

9) Please change "hours" to "hr" (lines 150, 306, 308, 313, 317, 350, 367, 417, 419, 426, 465, 468, 472, 473, 479, 545, 550, 672, 675, 757, 885, 888, 894, 895, 897, 899, 903, 905, 911, 918 2x, 935, 940, 942, 964, 966, 975, 984, 985, 986, 1008, 1010, 1063, 1064).

10) Please change "the induction of cell death and apoptosis using an antibody specific for PARP1 and Western Blot" to "in terms of its ability to induce apoptosis and cell death by Western Blot using an antibody specific for PARP1" or "in terms of its ability to induce apoptosis and cell death by Western Blot technique using an antibody specific for PARP1" (line 151).

11) Please specify the size of uncleaved PARP1 in kDa as part of "During caspase-dependent apoptosis, PARP1 is cleaved into 24 kDa and 89 152 kDa fragments by caspases 3 and 7" (line 152).

12) Please replace "establish" with something like "detect", "monitor", or "confirm" (line 154).

13) Please change "HCT116 p53- /-" to "HCT116 p53-/-" (line 156).

14) Please replace "cleavage" with "cleavage was observed" (line 157).

15) Please change "bands," to "bands" (line 158).

16) Please split "Further, detection of Annexin-V positivity, which marks apoptotic cells, in ND-Nic treated cells revealed a dose-dependent increase in drug-induced apoptosis in HCT116 p53-/- to a far greater extent when compared to HCT116 p53+/+ cells" into two sentences (line 160).

17) Please replace "HCT116 p53-/-" with "HCT116 p53-/- cells" (line 161).

18) The IC50 values (~2 to ~4 uM for niclosamide and ~5 to >15 for ND-Nic) mentioned in "It appeared that the removal of aniline 4’-NO2 group resulted in overall increased IC50 values in both p53+/+ and p53-/- cells compared to niclosamide (niclosamide ~2 to ~4 uM; ND-Nic ~5 to >15 uM)" (line 165), in "Approximate IC50 values are Nic (~1 to 2 μM) and ND-Nic (~7.5 to 16.5 μM)" (line 319), and in "The empirical IC50 values would however imply that ND-Nic may be a less potent drug (IC50 (ND-Nic) = ~7.5 to 16.5 μM compared to IC50 (Nic) = ~1 to 2 μM)" (line 838) seem not to correspond with what is shown in Figure 1H. Please provide more accurate dose ranges.

19) Please change "dose-effect" to "dose effect" (line 168).

20) It is not clear what the authors mean by "(X)" in The integral dose-response over the defined dose range (for example, between a and b) is calculated as a continuous function of the multiple parameters of the nonlinear dose-response curve (X) (line 177)? Please distinguish clearly from f(x).

21) Please split "The quantitative scoring of differential drug sensitivity between patient and control cells has been applied to optimise the selection of cancer-selective drugs as well as drug-sensitive patient sub-groups and outperforms other response parameters" into two sentences (line 180).

22) Please change "using cell proliferation" to "using" (line 184).

23) Please replace "following treatment with increasing doses of niclosamide or ND-Nic in HCT116 p53+/+ and p53-/- cells was measured" with "was measured following treatment with increasing doses of niclosamide or ND-Nic in HCT116 p53+/+ and p53-/- cells" (line 185).

24) The adjective "isogenic" seems to be redundant in "Areas under the individual dose-response curves and dDSS between isogenic HCT116 p53+/+ and p53-/- cells was calculated for each drug (refer to Materials and Methods) and shown in Figure 1H" (line 187).

25) Please change "was" to "were" (line 189).

26) Please replace "(refer to Materials and Methods) and shown in Figure 1H" with "and shown in Figure 1H (refer to Materials and Methods)" (line 189).

27) Please shorten "it is clear from the results that ND-Nic" to "ND-Nic" (line 190).

28) Please change "selectivity (for p53-/- cells)" to "selectivity for p53-/- cells" (line 199).

29) "Altogether, our data support the further characterisation of ND-Nic as a promising structural analogue that enhances the selective targeting of p53-deficient cells through its mitochondrial uncoupling function, and supports the investigation into potential nonspecific cytotoxicity that may accompany the inclusion of the aniline 4’-NO2 group on niclosamide" (line 202) is too long. Please split into at least two sentences.

30) It is not exactly clear what the authors mean by "data support the further characterisation" in "Altogether, our data support the further characterisation of ND-Nic as a promising structural analogue that enhances the selective targeting of p53-deficient cells through its mitochondrial uncoupling function, and supports the investigation into potential nonspecific cytotoxicity that may accompany the inclusion of the aniline 4’-NO2 group on niclosamide" (line 202)? 

31) Please split "Next, we sought to determine if ND-Nic retained the mitochondrial uncoupling function that is characteristic of niclosamide which is also the key mechanism driving synthetic lethality in p53-deficient cells" into two sentences (line 210).

32) Please replace "JC-1 dye which is a lipophilic, cationic dye" with "a lipophilic cationic dye JC-1" (line 212).

33) "When the mitochondrial membrane potential is diminished or abrogated, either by mitochondrial 216 uncoupling or in unhealthy apoptotic cells, the red/green fluorescence signal ratio is decreased, as displayed in the presence of the classical oxidative phosphorylation uncoupler, carbonyl cyanide 3-chlorophenylhydrazone (CCCP)" (line 215) is too long. Please split into at least two sentences.

34) Although the authors claim that "Similar to FCCP positive control, ND-Nic induced mitochondrial uncoupling as observed from the increase in the oxygen consumption rate (OCR) that is independent of oligomycin" (line 225), the dependency of the ND-Nic induced effect on oligomycin is not shown. Pleas provide data or rephrase the sentence accordingly.

35) Please change "revealed" to "revealed that there is" (line 228).

36) The adjective "isogenic" seems to be redundant in "A sharp drop in intracellular ATP concentration is observed with treatment with ND-Nic, similar to that seen with niclosamide in the isogenic HCT116 p53+/+ and p53-/- cells" (line 232).

37) Please replace "observed with" with "observed following" (line 233).

38) "These results are consistent with the observed depolarisation of mitochondrial membrane and indicated that despite the elimination of the aromatic nitro group in ND-Nic, its function in uncoupling mitochondrial oxidative phosphorylation is preserved, an observation that is also consistent with the conservation of its selective targeting of p53-deficient cancer cells" (line 234) is too long. Please split into at least two sentences.

39) Please change "No2" to "NO2" as part of the 4'-nitro aniline motif in Figure 1A.

40) Please provide x and y axis marks for both graphs presented in Figure 1C.

41) Please indicate how long were cells treated with DMSO, ND-Nic, and Nic in the legend to Figure 1E.

42) Please replace "Percentage (%) of apoptotic cells" to "Apoptotic cells (%)" in Figure 1F.

43) Please change "(Nic) - 5-chloro-N-(2-chloro-4-nitrophenyl)-2-hydroxybenzamide and" to "(Nic), 5-chloro-N-(2-chloro-4-nitrophenyl)-2-hydroxybenzamide, and" (line 303).

44) Please replace "(ND-Nic) - 5-chloro-303 N-(2-chlorophenyl)-2-hydroxy-benzamide" with "(ND-Nic), 5-chloro-303 N-(2-chlorophenyl)-2-hydroxy-benzamide" (line 303).

45) Please change "acts" to "was used" (lines 305, 310).

46) Please replace "post 7 days" with "7 d post" (lines 306, 549).

47) Please change "niclosamide (Nic) and nitro-deficient niclosamide (ND-Nic)" to "Nic and ND-Nic" (line 306).

48) Please change "Graph quantifies the colony intensity of the crystal violet staining results by ImageJ" to "Graph quantifies the colony intensity of the crystal violet staining results" (line 306).

49) Please replace "Brightfield" with "Bright field" (line 307).

50) Please change "nitro-deficient niclosamide (ND-Nic)" with something like "ND-Nic are shown" (line 308).

51) "Protein expression of apoptosis marker, cleaved PARP1 detection in either DMSO, 2 μM niclosamide (Nic) or 7.5 μM nitro-deficient niclosamide (ND-Nic) treated HCT116 p53+/+ and p53-/- cells detected by Western Blot" (line 308) seems not to be grammatically correct. Please revise.

52) Please change "niclosamide (Nic)" to "Nic" (lines 309, 316, 419, 425, 464, 467, 469, 476, 544, 757, 

53) Please replace "nitro-deficient niclosamide (ND-Nic)" with "ND-Nic" (lines 309, 419, 465, 470, 477, 545, 547, 671, 673, 681, 757).

54) Please replace "quantified are shown" with "are shown quantified" (line 311).

55) Please change "Annexin V" to "Annexin-V-Alexa Fluor 568" (line 311).

56) Please replace "0 μM, 5 μM, 7.5 μM, 10 μM" with "0, 5, 7.5, 10," (line 312).

57) Please change "ns = P > 0.05 and **** = P < 0.0001" to "****, P < 0.0001; ns, P > 0.05" (line 314).

58) Please replace "Drug Sensitive Score" with "drug sensitive score" (line 314).

59) Please change "of" to "of the" (line 315).

60) Please replace "plotted in the R program" with "plotted" (line 318).

61) Please change "curve, AUC" to "curve (AUC)" (line 318).

62) Please replace "damage, using" with "damage using" (line 323).

63) Please rephrase or split "Immunofluorescence detection of pRPAS4S8 under niclosamide treatment mirrored the trend observed in γH2AX whereby niclosamide treatment resulted in elevated pRPAS4S8 levels in a dose-dependent manner" (line 331) so that "niclosamide treatment" is mentioned only once per sentence.

64) From "Total RPA level detected using antibody against RPA70 showed a mild increase in RPA70 staining in response to niclosamide" (line 333) is not clear why RPA70 is an indicator of total RPA level?

65) Please change "indicated" to "indicate" (line 337).

66) Please replace "suggesting" with "suggesting that" (line 339).

67) "To further determine if niclosamide induces the formation of any DNA lesions, we employed the use of the comet assay which enables the measurement of DNA strand breaks at the cellular level based on the ability of broken DNA strands to migrate towards the anode under an electric field" (line 341) is too long. Please split into two sentences.

68) Please change "of any" to "of" (line 341).

69) Please replace "employed the use of" with "employed" or "deployed" (line 342).

70) "results in" could be changed to "gives rise to characteristic" (line 344).

71) Please change "migrating" to "migrating away" (line 345).

72) "using" could be changed to "using conventional" (line 345).

73) From "This parameter represents the product of the percentage of total DNA in the comet tail and the difference in the comet tail and head lengths" (line 346) is not clear how OTM is actually calculated. Would the authors mind providing a simple equation?

74) Please replace "olive tail moment" with "OTM" (lines 348, 1005).

75) Please change "(58) over the untreated and DMSO treated cells (Figure 3E)" to "over the untreated and DMSO treated cells (Figure 3E) (58)" (line 351).

76) Please replace "niclosamide treatment" with "niclosamide" (line 353).

77) "Micronuclei (MN) are recognised as small extra-nuclear bodies consisting of either damaged chromosome fragments from acentric chromatid/chromosome fragments or whole chromatids/chromosomes that trail behind replicated DNA during anaphase and signify chromosomal instability that results from mitotic errors or DNA damage" (line 358) is too long. Please split into at least two sentences.

78) Please change "Micronuclei (MN)" to "MN" (line 358).

79) Please replace "signify" with "indicate" (line 361).

80) Please change "risk and" to "risk and" (line 365).

81) Please replace "demonstrated that niclosamide induced" with "demonstrate that niclosamide induces" (line 365).

82) Please change "micronuclei" to "MN" (lines 366, 979, 981).

83) Please replace "Altogether the" with "Altogether, these" (line 369).

84) "More importantly, elimination of the aromatic nitro group was sufficient to attenuate DNA damage, as demonstrated using multiple molecular parameters to measure genotoxicity, suggesting that the aniline 4’-NO2 group is a major contributor to niclosamide-induced cellular DNA damage" (line 371) is too long. Please split into two sentences.

85) The second "+" in the "p53+/+" label as part of the quantification graph is slightly distorted/duplicated in Figure 2C. Please fix.

86) The "DMSO" label is cut through in the p53-/- quantification graph in Figure 2C. Please replot.

87) Please replace "Time (minutes)" to "Time (min)" in Figure 2D.

88) Please change "3) Antimycin A/rotenone" to "3: Antimycin A/rotenone" in Figure 2D.

89) Please replace "ATP Percentage (%)" with "ATP (%)" in both graphs presented as part of Figure 2F.

90) Please replace "7.5 μM" with "7.5" (line 419).

91) Please change "* = P < 0.05, ** = P < 0.01, *** = P < 0.001 and **** = P < 0.0001" to "*, P < 0.05; **, P < 0.01; ***, P < 0.001; ****, P < 0.0001" (line 420).

92) Please replace "( ns = P > 0.05)" with "(ns, P > 0.05)" (line 424).

93) Please change "Percentage (%)" to "Percentage" (lines 425, 478, 1016).

94) Please replace "(**** = P < 0.0001)" with "(****, P < 0.0001)" (lines 427, 466).

95) Please change "4 and" to "4, and" (lines 464, 473, 895).

96) Please replace "7.5 and" with "7.5, and" (lines 465, 473, 545, 757).

97) Please change "p53 and Hsp90" to "p53, and Hsp90" (line 467).

98) Please replace "Olive tail moment, OTM calculated by (tail length-head length) x % tail DNA" with "Olive tail moment (OTM), calculated as (tail length-head length) x % tail DNA," (line 467).

99) Please change "determine" to "determined" (line 485).

100) "We previously identified a potential mechanism underlying the selective killing of p53-deficient cells by niclosamide that is mediated through a newly characterised pathway involving concerted changes in calcium flux and phospholipid turnover that ultimately resulted in a lethal accumulation of arachidonic acid metabolites and PARP1/caspase-3 dependent apoptosis" (line 486) is too long. Please split into at least two sentences.

101) Please replace "Fluo-4, AM, dye that" with ""Fluo-4, AM dye, which" (line 501).

102) Please change "AM, dye" to "AM dye" (line 509).

103) Please replace "demonstrated" to "demonstrate" (line 515).

104) Please change "fluo-4" to "Fluo-4" (line 546).

105) Please replace "(ns = P > 0.05, * = P < 0.05, ** = P < 0.01 and *** = P < 0.001)" with "(*, P < 0.05; **, P < 0.01; ***, P < 0.001; ns, P > 0.05)" (line 548).

106) Please change "12 and" to "12, and" (lines 550, 896).

107) "Since niclosamide is known to induce differential metabolic changes in p53-deficient cancer cells, and cause the inhibition of p53-deficient cancer cells primarily through an arachidonic-acid dependent mechanism [18, 69, 70], we decided to investigate if ND-Nic may cause similar lipidomic changes in cells" (line 557) is too long. Please split into two sentences.

108) Please change "liquid-chromatography mass" to "liquid chromatography-mass" (line 675).

109) Please replace "(ns = P > 0.05, * = P < 0.05, ** = P < 0.01, *** = P < 0.001 and **** = P < 0.0001)" with "(*, P < 0.05; **, P < 0.01; ***, P < 0.001; ****, P < 0.0001; ns, P > 0.05)" (line 679).

110) Please change "ns = P > 0.05" to "ns, P > 0.05" (line 682).

111) Please replace "((20:5 (ω-690 3))" with "(20:5 (ω-690 3))" (line 690).

112) Please change "profiling results" to "profiling" (line 698).

113) "Similarly, we detected a significant induction of ALOX5 and ALOX12B gene transcripts in response to ND-Nic and a greater extent in p53+/+ compared to p53-/- cells (Figure 6B), suggesting that the induction of ALOX5 and ALOX12B genes is dependent on p53, a notion that is consistent with the previously reported observation in niclosamide studies and also in Figure 5B" (line 720) is too long. Please split into at least two sentences.

114) Please replace "critical to" to "critical for" (line 727).

115) "Altogether, our results suggest that ND-Nic functionally phenocopies the cellular effects of niclosamide, with corroborating evidence that ND-Nic induces similar lipidomic and transcriptional changes, and parallels the action of niclosamide in eliminating p53-deficient cancer cells through a calcium-dependent pathway" (line 728) is too long. Please split into two sentences.

116) Please change "2, and" to "2 and" (line 757).

117) Please replace "(ns = P > 0.05, * = P < 0.05, ** = P 758 < 0.01, *** = P < 0.001 and **** = P < 0.0001)" with "(*, P < 0.05; **, P < 0.01; ***, P < 0.001; ****, P < 0.0001; ns, P > 0.05)" (line 758).

118) "While most studies have focused on elucidating structural changes that either enhance or compromise the effects of niclosamide on tumour cell killing and correlating that to its protonophoric activity, we found that niclosamide induces pleiotropic effects on DNA damage which is directed through its aniline 4’-NO2 group" (line 764) is too long. Please split into two sentences.

119) "The separation of the structure-function relationship between its antitumour mechanism driven by the phenolic hydroxyl group promoting mitochondrial uncoupling and the nonspecific genotoxicity imposed by its nitro group on the aniline motif, suggests that the two outcomes may be functionally uncoupled" (line 768) is too long. Please split into two sentences.

120) Please change "re-examine" to "re-examined" (line 819).

121) "arachidonic acid" could be replaced with "Ca2+/AA" (line 823).

122) The adjective "isogenic" seems to be redundant in "Using quantitative measurements of differential drug sensitivity scores (dDSS) in the isogenic human colorectal HCT116 p53+/+ and p53-/- cancer cells, we compared the differential drug response" (line 823).

123) Please change "cells, using HCT116 p53+/+ as the control cells" to "cells" (line 831).

124) Please replace "37°C" with "37 °C" (lines 881, 890, 914, 991, 1012).

125) Please change "days" to "d" (lines 889, 898).

126) Please replace "440nm" with "440 nm" (line 891).

127) Please change "10 and" to "10, and" (line 896).

128) Please replace "and were" with "and" (line 898).

129) Please change "hours respectively" to "hours" (line 905).

130) Please replace "minutes" with "min" (lines 914, 933, 938, 947, 968, 969, 970, 971, 973, 976, 994, 995, 996, 997, 998, 999 2x, 1001, 1012, 1068).

131) Please change "concentrations" to "concentration" (line 925).

132) Please replace "4°C" with "4 °C" (lines 928, 973, 995 2x, 996, 998).

133) Please change "A300-245A ;" to "A300-245A;" (line 931).

134) Please specify the type of Tween used in "PVDF membrane was subsequently washed for 10 minutes in 1x PBS with 0.1% Tween for 3 times prior to incubation in horseradish peroxidase-conjugated secondary antibodies (Agilent Dako; P0161, P0448; 1:5000) for 2 hours at room temperature" (line 932) and in "ECL detection reagent (Cytiva Amersham 45-00-999) prepared as recommended by the manufacturer’s protocol was used for the visualisation of specific protein expression on X-ray film after 3 times of 10 minutes washes in 1x PBS with 0.1% Tween" (line 935).

135) Please change "and were" to "and" (line 940).

136) Please replace "5 μM, 7.5 μM and" with "5, 7.5, and" (lines 941, 965, 985).

137) Please change "95°C" to "95 °C" (line 947 2x).

138) Please replace "60°C" with "60 °C" (line 948).

139) Please change "72°C" to "72 °C" (line 948).

140) Please replace "2 μM, 4 μM and" with "2, 4, and" (lines 965, 985).

141) Please change "fixing them" to "fixation" (line 966).

142) Please replace "main nuclei" with "nucleus" (line 980).

143) Please change "in" to "with" (line 984).

144) Please replace "and was" with "and" (line 994).

145) Please change "Green" to "green" (line 1000).

146) Please replace "−80°C" with "−80 °C" (line 1043).

147) Please define abbreviation for "LC-MS" (line 1045).

148) Please change "pH7.4" to "pH 7.4" (line 1056).

149) From "4.13 Mitochondrial cellular respiration" section is not clear at what temperature were oxygen consumption rates measured in the Seahorse assay?

150) Please replace "Annexin-V-Alexa" with "Annexin-V-Alexa Fluor" (line 1066).

151) Please change "ROCHE" to "Roche" (line 1067 2x).

Author Response

Dear Reviewers, 

We thank you for your time in reviewing our manuscript and providing very helpful comments in both rounds of the revision. We have edited carefully and accordingly to the suggestions. We provide a point-by-point response to the suggestions in the attachment.  We hope this has addressed all the comments that has substantially improved our manuscript.

Sincerely,

Chit Fang Cheok

Round 3

Reviewer 2 Report

1) "would provide an optimal scenario for drug design" and "with future development focused on optimising drug design" seem to be redundant with respect to each other in "Ideally, the functional uncoupling between the defined biological activity of a drug from any pleiotropic effects or nonspecific toxicity in structure-activity experiments would provide an optimal scenario for drug design, with future development focused on optimising drug design through reducing or eliminating nonspecific drug action" (line 54).

2) Please change "action mechanism" to "mechanism of action" (line 104).

3) Please replace "lipidomic" to "lipidomics" (lines 105, 567, 570, 696, 816, 821).

4) Please change "demonstrating that collateral DNA damage accompanying niclosamide may be attenuated in ND-Nic supports" to "demonstrate that collateral DNA damage accompanying niclosamide may be attenuated with ND-Nic and support" (line 106).

5) It is not clear what the authors mean by "synthetic lethal inhibition of p53-deficient cancer cells" in "Previously, we had identified niclosamide from a library screen of FDA-approved compounds in their capacity to target p53-deficient cells, causing synthetic lethal inhibition of p53-deficient cancer cells and tumour xenografts mediated through mitochondrial uncoupling" (line 112)?

6) Please change "24 kDa and 89" to "89 kDa and 24" (line 153).

7) There is the same problem as in the last revision round in that "Nic (~1 to 2 μM) and ND-Nic (~7.5 to 16.5 μM)" values do not exactly correspond with Figure 1H in "It appeared that the removal of aniline 4’-NO2 group resulted in overall increased IC50 values in both p53+/+ and p53-/- cells compared to niclosamide (Nic ~1 to 2 μM; ND-Nic ~7.5 to 16.5 μM)" (line 166), "Approximate IC50 values are Nic (~1 to 2 μM) and ND-Nic (~7.5 to 16.5 μM)" (line 319), and "The empirical IC50 values would however imply that ND-Nic may be a less potent drug (IC50 (ND-Nic) = ~7.5 to 16.5 μM compared to IC50 (Nic) = ~1 to 2 μM)" (line 836). Please provide more precise values in these sentences so that when looking from the 50 % Cell inhibition on the y axis intercepting each sigmoidal graph fit at a particular point (y intercept), the intercept from this point toward the x axis can be read as the Nic or ND-Nic IC50 concentration (x intercept).

8) Please describe explicitly which "Nic (~1 to 2 μM) and ND-Nic (~7.5 to 16.5 μM)" values refer to p53+/+ and which to p53-/- cells using the word "respectively" in "It appeared that the removal of aniline 4’-NO2 group resulted in overall increased IC50 values in both p53+/+ and p53-/- cells compared to niclosamide (Nic ~1 to 2 μM; ND-Nic ~7.5 to 16.5 μM)" (line 166), "Approximate IC50 values are Nic (~1 to 2 μM) and ND-Nic (~7.5 to 16.5 μM)" (line 319), and "The empirical IC50 values would however imply that ND-Nic may be a less potent drug (IC50 (ND-Nic) = ~7.5 to 16.5 μM compared to IC50 (Nic) = ~1 to 2 μM)" (line 836).

9) Please change "WST1" to "WST-1" (lines 186, 317).

10) Please replace "genotoxicity in" with "genotoxicity of" (line 358).

11) Please change "demonstrate" to "demonstrated" (lines 367, 514).

12) Please replace "as" with "with" (line 414).

13) Please change "(48 hr) treated HCT116 cells" to "treated HCT116 cells for 48 hr" (line 427).

14) Please change "( (D)" to "(D)" (line 466).

15) Please replace "Inset: a" with "Bottom: A" (line 469) as the image is not inserted as an inset.

16) Please change "cells" to "cells for 48 hr" (line 475).

17) Please change "Inset: white arrow indicates a micronucleus formed adjacent to a parent nucleus. Bottom:" to "Bottom left: White arrow indicates a micronucleus formed adjacent to a parent nucleus. Bottom right:" (line 476) as the image is not inserted as an inset.

18) Please replace "determined" with "aimed to determine" (line 484).

19) It is not clear from the legend to Figures 5B,D how long were cells treated with 7.5 μM ND-Nic?

20) Please change "(16 hr) treated human colon carcinoma HCT116 p53+/+ and p53-/- cells" to "treated human colon 670 carcinoma HCT116 p53+/+ and p53-/- cells for 16 hr" (line 670).

21) Please replace "(16 hr) treated HCT116 p53-/- cells" with "treated HCT116 p53-/- cells for 16 hr" (line 673).

22) Please change "liquid chromatography-mass spectrometry" to "liquid chromatography-mass spectrometry (LC-MS)" (line 562) and "liquid chromatography-mass spectrometry" to "LC-MS" (lines 673, 1039, 1040).

23) Please change "for to a" to "for a" (line 725).

24) Please replace "transcripts" with "transcript" (line 754).

25) Please replace "WST" with "WST-1" (line 887).

26) Please change "and were incubated" with "and cells were incubated with WST" (line 888).

27) Please replace "nitro-deficient niclosamide" with "ND-Nic" (lines 892, 901, 908, 915, 938, 1005, 1059).

28) Please change "and were" to "and" (line 895).

29) Please replace "5 μM, 7.5 μM" with "5, 7.5," (line 908).

30) Please change "deionised (DI)" to "deionised" (line 994).

31) Please change "2 μM" to "2" (lines 914, 937, 1004).

32) Please replace "7.5 μM" with "7.5" (lines 914, 1005).

33) Please change "XF96 well" to "XF96" (line 1048).

34) The dosage of FCCP is not clear from the "4.13. Mitochondrial cellular respiration" section. Was this a titration? If so, how many additions of FCCP were made and does 0.5 uM refer to one single addition or final concentration following titration?

35) Please replace "for" with "to" (line 1064).

36) Please change "Y.H." to "Y.S.H." (lines 1076, 1077, 1078, 1079, 1080).

37) Please replace "G.E." to "G.A.E." (lines 1077 2x, 1078, 1079 2x, 1080).

38) Please change "Y.H" to "Y.S.H." (line 1078).

39) Please replace "C.F.C" with "C.F.C." (line 1078).

40) Please change "Ken" to "Ken," (line 1088).

Author Response

Dear Editor and reviewers,

Thank you for your comments and suggestions in the third round of revision. We have now edited the manuscript accordingly and hope we have addressed all areas of concern. We look forward to working with you on the successful publication at IJMS.

Sincerely,
Chit Fang
